# *Drosulfakinin* signaling encodes early-life memory for adaptive social plasticity

Jiwon Jeong[1†], Kujin Kwon[2†], Terezia Klaudia Geisseova[1], Jongbin Lee[3], Taejoon Kwon[2,4,5]*, Chunghun Lim[3,6,7]*

[1]Department of Biological Sciences, Ulsan National Institute of Science and Technology, Ulsan, Republic of Korea; [2]Department of Biomedical Engineering, Ulsan National Institute of Science and Technology, Ulsan, Republic of Korea; [3]Research Center for Cellular Identity, Korea Advanced Institute of Science and Technology, Daejeon, Republic of Korea; [4]Center for Genomic Integrity, Institute for Basic Science, Ulsan, Republic of Korea; [5]Graduate School of Health Science and Technology, Ulsan National Institute of Science and Technology, Ulsan, Republic of Korea; [6]Department of Biological Sciences, Korea Advanced Institute of Science and Technology, Daejeon, Republic of Korea; [7]Graduate School of Stem Cell and Regenerative Biology, Korea Advanced Institute of Science and Technology, Daejeon, Republic of Korea

*For correspondence:
tkwon@unist.ac.kr (TK);
clim@kaist.ac.kr (CL)

[†] These authors contributed equally

Competing interest: The authors declare that no competing interests exist.

**Abstract** *Drosophila* establishes social clusters in groups, yet the underlying principles remain poorly understood. Here, we performed a systemic analysis of social network behavior (SNB) that quantifies individual social distance (SD) in a group over time. The SNB assessment in 175 inbred strains from the *Drosophila* Genetics Reference Panel showed a tight association of short SD with long developmental time, low food intake, and hypoactivity. The developmental inferiority in short-SD individuals was compensated by their group culturing. By contrast, developmental isolation silenced the beneficial effects of social interactions in adults and blunted the plasticity of SNB under physiological challenges. Transcriptome analyses revealed genetic diversity for SD traits, whereas social isolation reprogrammed select genetic pathways, regardless of SD phenotypes. In particular, social deprivation suppressed the expression of the neuropeptide Drosulfakinin (*Dsk*) in three pairs of adult brain neurons. Male-specific DSK signaling to cholecystokinin-like receptor 17D1 mediated the SNB plasticity. In fact, transgenic manipulations of the DSK neuron activity were sufficient to imitate the state of social experience. Given the functional conservation of mammalian *Dsk* homologs, we propose that animals may have evolved a dedicated neural mechanism to encode early-life experience and transform group properties adaptively.

## Editor's evaluation

This study presents important findings on the role of Drosulfakinin signaling in encoding early-life social memory in *Drosophila*, which influences adaptive social plasticity in adulthood. The research demonstrates the neurogenetic basis of social clustering and behavioral adaptation, advancing our understanding of social behavior's molecular and evolutionary bases. The evidence is solid, given the robust genetic, behavioral, and transcriptomic analyses.

## Introduction

Animals interact with other individuals in distinct social environments (*Clutton-Brock, 2021*; *Jezovit et al., 2021*; *Sokolowski, 2010*). For instance, a pair of animals display aggression or mating

behaviors, whereas a group of individuals may show collective behaviors through intricate networks of social interactions. Such social network behaviors (SNBs) are believed to enhance group fitness and are conserved across many species, underscoring their evolutionary significance (*Sokolowski, 2010*; *Blumstein et al., 2010*). Additionally, social interactions influence various physiological activities in individuals, including feeding, sleep/circadian rhythms, aggression, stress response, and longevity (*Levine et al., 2002*; *Ganguly-Fitzgerald et al., 2006*; *Arzate-Mejía et al., 2020*; *Li et al., 2021*; *Chen and Sokolowski, 2022*; *Vora et al., 2022*; *Xiong et al., 2023*). Nonetheless, it remains elusive how group properties have evolved with other individual traits and how animals process social experiences to shape their behavior and physiology.

*Drosophila* has long been considered a solitary species. Yet a group of flies can display distinct SNB under specific experimental conditions (*Schneider et al., 2012*; *Simon et al., 2012*; *Ramdya et al., 2015*; *Dombrovski et al., 2017*; *Sun et al., 2020*; *Burg et al., 2013*), serving as an ideal genetic model to address our questions above (*Jezovit et al., 2021*). Previous studies have employed various biophysical parameters to quantify both individual and network behaviors in groups, establishing criteria for social interactions in *Drosophila* (*Jezovit et al., 2021*; *Simon et al., 2012*; *Ramdya et al., 2015*; *Bentzur et al., 2021*). While these measurements give a comprehensive view of group behaviors, our study focuses on the clustering property of social interactions within a group (*Simon et al., 2012*; *Jiang et al., 2020*). Social clustering is an intuitive measure that integrates diverse social interactions via multiple sensory cues (*Schneider et al., 2012*; *Simon et al., 2012*; *Bentzur et al., 2021*; *Jiang et al., 2020*), accompanying reductions in social distance (SD) among group members and their moving speed over time. We also reasoned that simple SD assessment could facilitate the alignment of large-scale datasets of group behaviors with physiological traits, differential gene expression, and neurogenetic manipulations.

These approaches lead to our demonstration that the group property for high social clustering (i.e., short SD) is closely associated with and compensates for inferior developmental traits in individuals. Moreover, *Drosophila* can adjust their social preferences according to physiological changes, indicating the adaptive plasticity of SNB. This social plasticity requires early-life social experiences that persist throughout development. We also define a specific neuropeptide signaling pathway that encodes social memory and supports SNB plasticity. Our findings thus provide new insights into the principles of SNB, offering a deeper understanding of the evolutionary and genetic bases of social behaviors in *Drosophila* and possibly other species.

## Results

### SNB is a quantitative trait in a natural *Drosophila* population

We employed the *Drosophila* Genetics Reference Panel (DGRP) to determine whether SNB has evolved with specific genetic factors and physiology. The DGRP consists of approximately 200 inbred wild-type strains and functions as a practical genetic library to explore the correlation between naturally occurring genetic variations and complex animal behaviors (*Mackay et al., 2012*; *Mackay and Huang, 2018*; *Gardeux et al., 2024*). We video-recorded a group of 16 male flies freely moving in an open arena for 10 min and quantitatively assessed their group properties over time (*Figure 1A*). These included SD between individual group members (*Figure 1B*), walking speed, and the centroid velocity of a given group. The DGRP lines displayed a range of distributions for the three parameters (*Figure 1C*, *Figure 1—source data 1*), and we found their significant correlations among 175 DGRP lines (*Figure 1D*, *Figure 1—source data 1*; Spearman correlation analysis). For instance, short-SD lines gradually reduced SD and walking speed over time to stay in the cluster, thereby exhibiting short travel distances, slow walking speeds, and low centroid velocities on average (e.g., DGRP73, DGRP 563, and DGRP370) (*Figure 2A*, *Figure 2—figure supplement 1*, *Figure 2—source data 1*, *Video 1*). By contrast, long-SD lines persistently explored the arena during video recording and sustained 'social distancing' to display long travel distances, fast walking speeds, and high centroid velocities (e.g., DGRP360, DGRP707, and DGRP317) (*Figure 2B*, *Figure 2—figure supplement 1*, *Figure 2—source data 1*, *Video 2*), although their locomotion was modestly slowed down over time. The locomotion trajectories of individual flies confirmed these characteristics (*Figure 2C and D*, *Figure 2—source data 1*), and long-SD individuals traveled much longer distances than short-SD individuals across the DGRP lines (*Figure 2—figure supplement 1C*, *Figure 2—source data 1*). Short-SD flies did

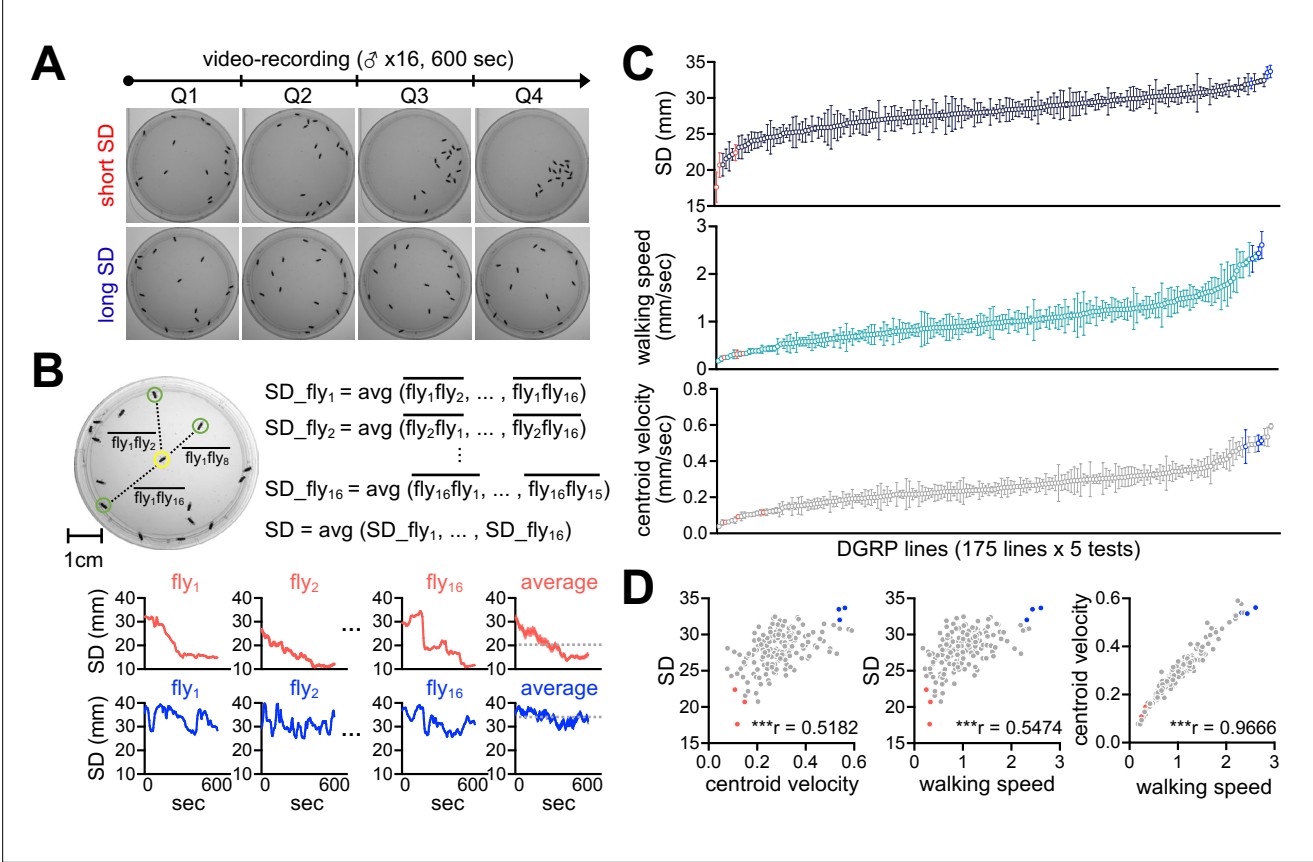

**Figure 1.** Social network behavior (SNB) is a quantitative trait in a natural *Drosophila* population. (**A**) The 10 min video recording of SNB in a group of 16 male flies. Representative snapshot images at each quarter time point (Q1, Q2, Q3, and Q4) were obtained from *Drosophila* strains with high (top, short social distance [SD]) or low clustering properties (bottom, long SD). (**B**) The definition of SD. SD was measured in each fly over the 10 min recording and averaged from a given group. Representative SD dynamics from short- (top) or long-SD strains (bottom) were shown. Dotted lines, group-averaged SD over the 10 min recordings. (**C**) Quantitative assessment of SNB by ranking SD, walking speed, and centroid velocity among 175 DGRP lines. Data represent means ± SEM (n = 5). (**D**) Significant correlation among SD, walking speed, and centroid velocity. ***p<0.001, as determined by Spearman correlation analysis. Red, representative short-SD lines; blue, representative long-SD lines.

The online version of this article includes the following source data for figure 1:

**Source data 1.** Correlation of SD, walking speed, and centroid velocity across 175 DGRP lines.

not significantly change their locomotor activity over time when we placed a single fly in the same arena and assessed its behavior (*Figure 2—figure supplement 2*, *Figure 2—source data 1*). Thus, reduced activity in a group of short-SD flies is likely an effect of their clustering phenotypes but not necessarily the cause. Short- and long-SD flies retained their clustering properties even in a larger arena (*Figure 2—figure supplement 3*, *Videos 3 and 4*; 8.5 cm in diameter), indicating active social preferences that persist in different environments. Both types of DGRP lines displayed no chaining behaviors in the open arena, excluding the possible implication of male-to-male courtship in their SD phenotypes (*Figure 2—figure supplement 4*, *Figure 2—source data 1*). These results provide convincing evidence that SD is an inheritable group trait from natural *Drosophila* variants. Based on the average ranking of individual DGRP lines in the two group properties (i.e., SD and centroid velocity), we selected the top and bottom three DGRP lines representing short- and long-SD phenotypes, respectively, to elucidate the physiological significance of *Drosophila* SNB and its underlying mechanisms.

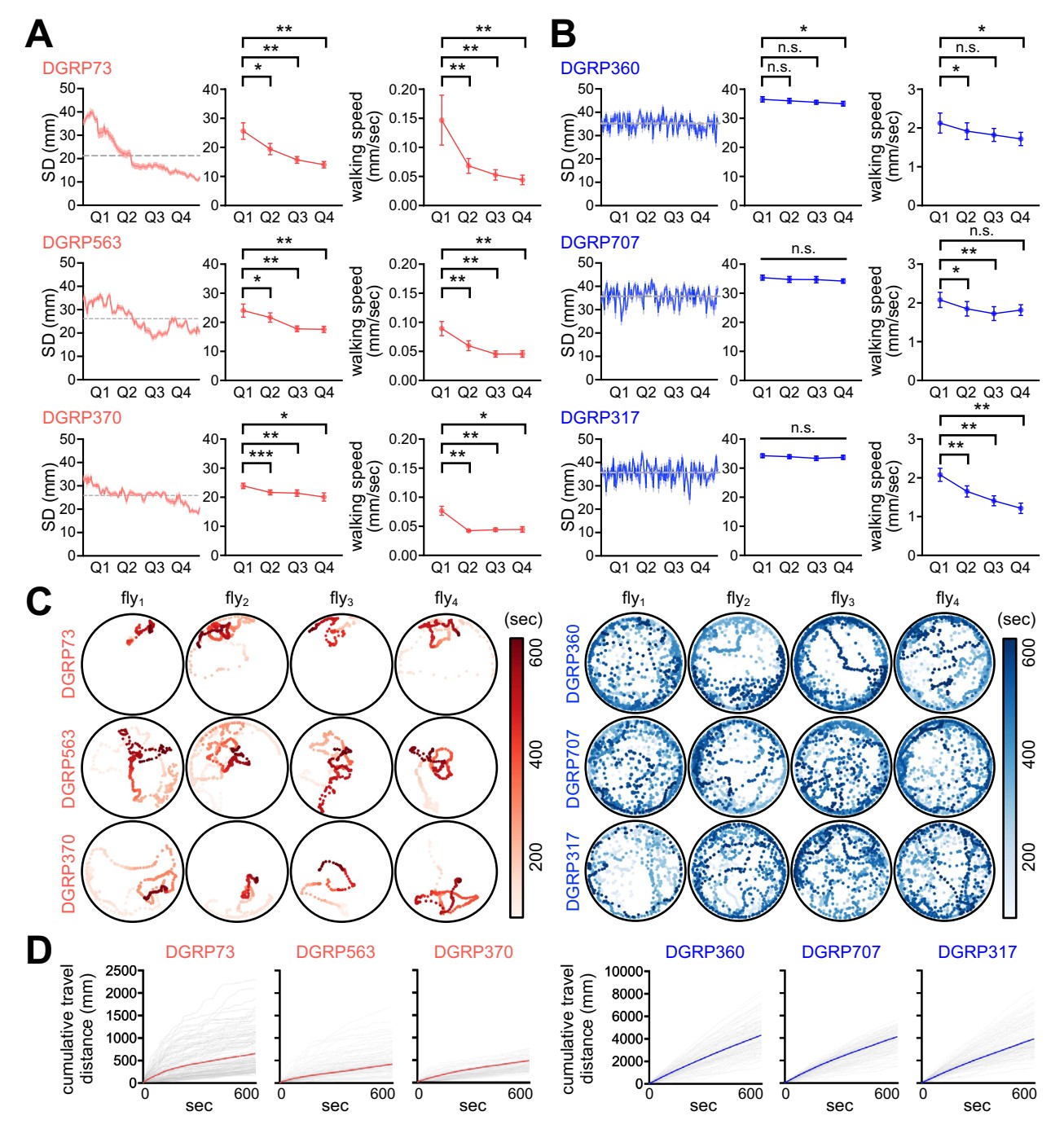

**Figure 2.** Short- and long-social distance (SD) lines exhibit distinct social network behavior (SNB). (**A, B**) SNB dynamics in short-SD (**A**) and long-SD lines (**B**). SD dynamics in a representative group of 16 male flies over the 10 min recordings (left, n = 16), quarter-averaged SD (middle, n = 8), and quarter-averaged walking speed (right, n = 8) were shown for each *Drosophila* Genetics Reference Panel (DGRP) line. Error bars indicate SEM. n.s., not significant; *p<0.05, **p<0.01, ***p<0.001 as determined by paired *t*-test (DGRP73, DGRP563, DGRP370, DGRP707, and DGRP317 for quarter-averaged SD; DGRP563, DGRP360, and DGRP317 for quarter-averaged walking speed) or Wilcoxon matched-pairs signed rank test (DGRP360 for quarter-averaged SD; DGRP73, DGRP370, and DGRP707 for quarter-averaged walking speed). (**C**) The 10 min locomotion trajectories of representative individual flies from short- (red) or long-SD lines (blue). (**D**) Cumulative travel distances of individual flies over the 10 min recording. Colored lines represent means (n = 128).

The online version of this article includes the following source data and figure supplement(s) for figure 2:

**Source data 1.** Quantitative locomotor metrics in short- and long-SD DGRP lines.

*Figure 2 continued on next page*

*Figure 2 continued*

**Figure supplement 1.** Social network behaviors in the three representative *Drosophila* Genetics Reference Panel (DGRP) lines displaying short or long social distance (SD).

**Figure supplement 2.** Neither short- nor long-social distance (SD) lines reduce their walking speeds over time when individual flies from group cultures are isolated.

**Figure supplement 3.** The clustering property of each *Drosophila* Genetics Reference Panel (DGRP) line persists in a large arena.

**Figure supplement 4.** Neither short- or long-social distance (SD) lines display male–male courtship behavior.

## Social interactions compensate for developmental inferiority in short-SD larvae

Why do flies display SNB? One clue comes from the previous observation that *Drosophila* larvae collectively dig culture media and improve food accessibility, possibly facilitating their constitutive feeding during early development (*Dombrovski et al., 2017*). In fact, we found that the short-SD lines had higher numbers of larvae per cluster than the long-SD lines (*Figure 3A*, *Figure 3—source data 1*). These observations suggest that *Drosophila* express SNB traits from early development, and the sociality persists in adults. To better define the social-interaction effects on *Drosophila* physiology, we obtained socially enriched or deprived larvae from fertilized eggs (*Figure 3B*) and then compared their developmental phenotypes among DGRP lines. Social isolation substantially impaired larval activity and development in both short- and long-SD lines (*Figure 3C–G*, *Figure 3—source data 1*). We further found that the short-SD trait is tightly associated with inferior phenotypes, particularly in isolated larvae. For instance, socially isolated larvae from the short-SD lines displayed poor digging activity (*Figure 3C*, *Figure 3—source data 1*), low food intake (*Figure 3D*, *Figure 3—source data 1*), and long developmental time (*Figure 3E*, *Figure 3—source data 1*) compared to those from the long-SD lines. Short-SD larvae also had lower eclosion success (*Figure 3F*, *Figure 3—source data 1*) and a lower ratio of the adult male progeny (*Figure 3G*, *Figure 3—source data 1*) than long-SD larvae, although isolated cultures reduced the eclosion success regardless of the SD trait. Grouping of short-SD larvae blunted or partially rescued these phenotypes (i.e., digging activity, developmental time, eclosion success, and male progeny ratio). Our group culture was not a competitive environment for limited resources to delay developmental time (*Horváth and Kalinka, 2016*; *Klepsatel et al., 2018*), but it actually promoted food intake comparably in short- and long-SD larvae (*Figure 3D*, *Figure 3—source data 1*). The significant interaction effects of SD (i.e., short- vs. long-SD trait) and socialization (i.e., grouped vs. isolated cultures) on digging activity, developmental time, and male progeny ratio suggest that the clustering property of short-SD lines may have evolved as

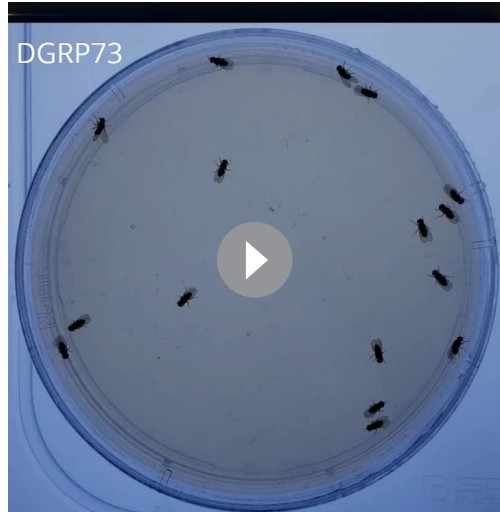

**Video 1.** Social network behavior (SNB) in DGRP73 (grp+ctrl).
https://elifesciences.org/articles/103973/figures#video1

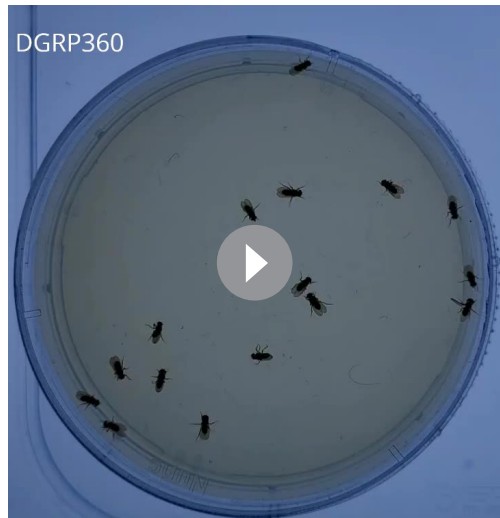

**Video 2.** Social network behavior (SNB) in DGRP360 (grp+ctrl).
https://elifesciences.org/articles/103973/figures#video2

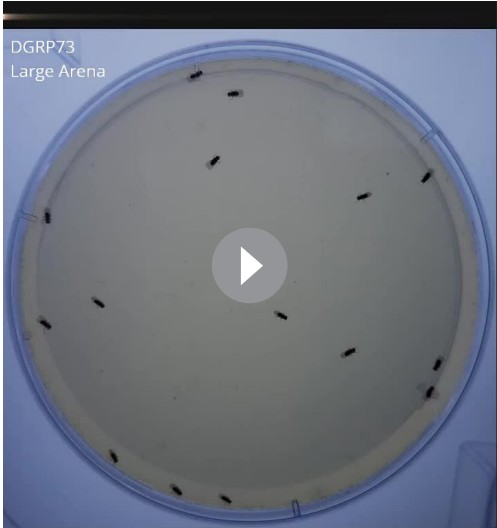

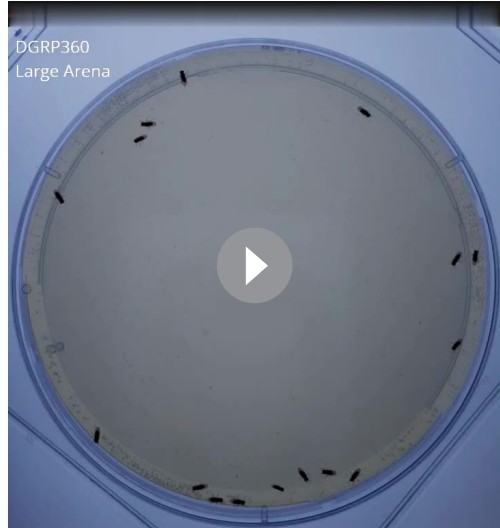

**Video 3.** Social network behavior (SNB) in DGRP73 (large arena).
https://elifesciences.org/articles/103973/figures#video3

**Video 4.** Social network behavior (SNB) in DGRP360 (large arena).
https://elifesciences.org/articles/103973/figures#video4

a compensation mechanism for the developmental inferiority in individuals. Considering that a low percentage of male likely limits mating choice, reproductive efficiency, and genetic diversity in a given group, social interactions may also contribute to the group fitness in short-SD lines across generations. We speculate that the feeding amount of isolated long-SD individuals is saturating for normal development (e.g., developmental time, male progeny ratio), possibly explaining the lack of interaction effects on food intake while displaying developmental inferiorities most evidently in isolated short-SD individuals.

## Early-life experience is necessary for social benefits on adult physiology and adaptive social plasticity

We further asked whether social interactions also benefit adult physiology in the short-SD lines. To this end, we designed a maze experiment where a group of flies were placed in a novel arena to determine how fast they could reach a food resource in the presence or absence of pretrained flies (*Figure 4A*). Our prediction was that social interactions between naïve and trained flies might facilitate their food-seeking, possibly mimicking social foraging in other species (*Giraldeau and Caraco, 2000*). We first confirmed that both short- and long-SD lines significantly shortened latency to the arrival of 75% of flies on food by iterative exposures to the maze (*Figure 4B*, *Figure 4—figure supplement 1*, *Figure 4—source data 1*). Representative plasticity mutants of the *rutabaga* (*rut*) gene did not significantly shorten the arrival latency after three consecutive training sessions in our maze paradigm (*Figure 4—figure supplement 2A*, *Figure 4—source data 1*), implicating *rut*-dependent learning and memory in this process (*Levin et al., 1992*). The short-SD lines displayed poor performance in the maze assay as assessed by longer latency to the arrival of 75% naïve flies on food than the long-SD lines (*Figure 4B*, *Figure 4—figure supplement 1*, *Figure 4—source data 1*). Combining the trained 'pioneers' with a group of naïve flies significantly improved the group property of food-seeking behaviors in both short- and long-SD lines (*Figure 4B*, *Figure 4—figure supplement 1*, *Figure 4—source data 1*). The pioneer effects disappeared when the naïve group consisted of socially isolated individuals from egg development. These results support that social foraging in our maze paradigm is unlikely a simple collective response in adults, but it specifically requires early-life social experience. Of note, social deprivation effects on pioneer-free group foraging somewhat varied across the long SD lines (*Figure 4B*, *Figure 4—figure supplement 1*, *Figure 4—source data 1*). We reason that the hyperactivity of individual long-SD flies facilitates their food-seeking behaviors in the maze, weakening the pioneer effects or even overriding the group property.

The long-SD lines outcompeted the short-SD lines in both larval and adult assays, and the beneficial effects of their social experience were barely detectable. SNB in the long-SD adults was also

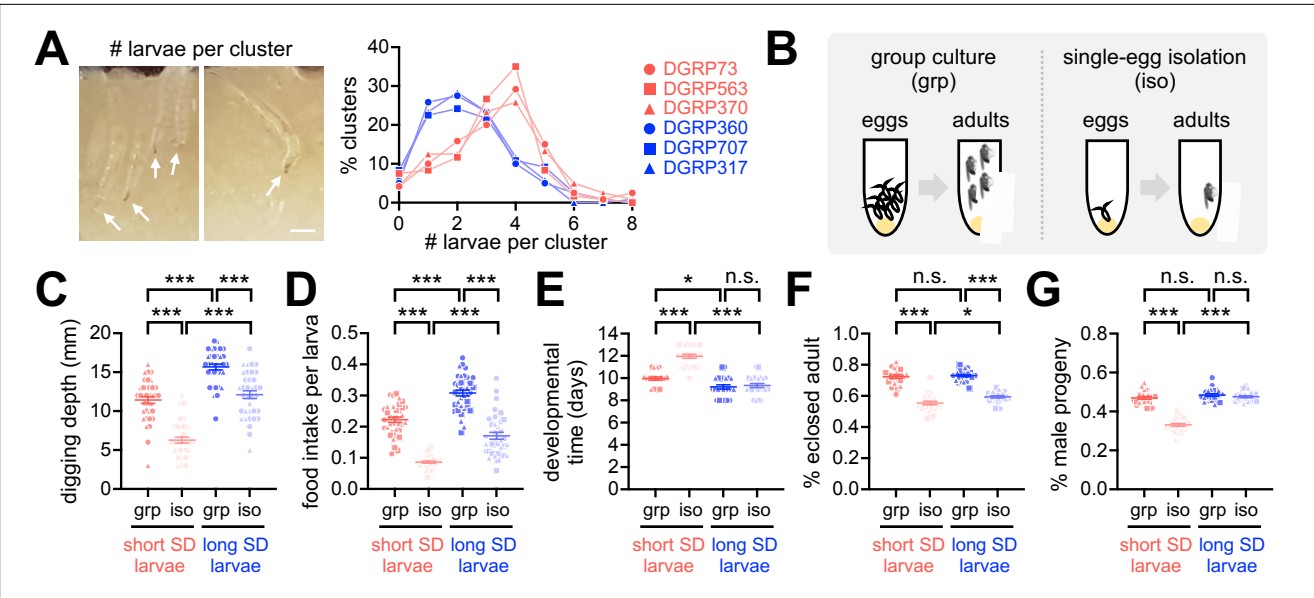

**Figure 3.** Early-life social experience confers beneficial effects on *Drosophila* development. (**A**) Larval clustering in short- (red) or long-social distance (SD) lines (blue). % clusters were calculated from 30 vials. Arrows indicate individual larvae. Scale bar = 0.5 mm. (**B**) Schematic for grouped (grp) vs. developmentally isolated (iso) culture conditions. (**C, D**) Grouped culture compensated for low food accessibility in the short-SD larvae. Aligned ranks transformation ANOVA detected significant effects of SD trait and social isolation on both digging depth (**C**, p<0.0001) and food intake (**D**, p<0.0001), and their significant interaction effects were detected only on digging depth (**C**, p=0.0338). Data represent means ± SEM (n = 36; 12 per line × 3 lines). ***p<0.001, as determined by Wilcoxon rank sum test. (**E–G**) Grouped culture rescued developmental delay and low male-progeny ratio in the short-SD larvae. Aligned ranks transformation ANOVA or ordinary two-way ANOVA detected significant effects of SD trait and social isolation on developmental time (**E**, p<0.0001), eclosion success (**F**, p=0.0116 for SD trait; p<0.0001 for social isolation), and male progeny ratio (**G**, p<0.0001). Significant interaction effects between SD trait and social isolation were also detected on developmental time (**E**, p<0.0001) and male progeny ratio (**G**, p<0.0001). Data represent means ± SEM (n = 24; 8 per line × 3 lines). n.s., not significant; *p<0.05, ***p<0.001, as determined by Wilcoxon rank sum test (developmental time and male progeny ratio) or Tukey's multiple comparisons test (eclosion success).

The online version of this article includes the following source data for figure 3:

**Source data 1.** Quantitative analysis of larval SNB and developmental metrics.

insensitive to early-life experience, whereas the grouped culture promoted SNB in the short-SD lines (*Bentzur et al., 2021*; *Figure 2—figure supplement 1B*, *Figure 2—source data 1*). We hypothesized that superior traits in long-SD individuals led to their genetic selection toward the degeneration of social interaction effects. Alternatively, long-SD genomes may still encode genetic programs for social activity, but they only express social traits under physiologically challenging conditions. Given the positive correlation between locomotion activity and SD trait among DGRP lines (*Figure 1D*, *Figure 1—source data 1*), we reasoned that modest injury could serve as a physiological cue to reduce locomotor activity in individuals, facilitate their interactions in a group, and induce SNB even in the long-SD flies. We indeed discovered that mechanical injury shortened SD in both sociality types of DGRP lines (*Figure 4C and D*, *Figure 4—figure supplement 3A*, *Figure 4—source data 1*; *Videos 5 and 6*;) while reducing walking speed and centroid velocity only in the long-SD lines (*Figure 4—figure supplement 4*, *Figure 4—source data 1*). The injury effects were transient because 1 week of recovery was sufficient to restore the original SD traits (*Figure 4D*, *Figure 4—figure supplement 3A*, *Figure 4—source data 1*). The injury-induced plasticity of SNB and locomotor activity was not detectable in a group of socially isolated flies (*Figure 4D*, *Figure 4—figure supplements 3A and 4*, *Figure 4—source data 1*, *Videos 7–10*). We thus reason that the mechanical injury does not severely impair general locomotion per se to abolish or overestimate SNB under our experimental conditions, but low activity in grouped long-SD flies is likely a consequence of their injury-induced clustering. The injury-induced SNB plasticity required *no receptor potential A* (*norpA*) among sensory pathway genes tested (*Figure 4—figure supplement 5*, *Figure 4—source data 1*), indicating a crucial role of *norpA*-dependent visual sensing. It is consistent with the previous finding that vision is required for larval clustering behaviors in *Drosophila* (*Dombrovski et al., 2017*). We further found that an adult-specific

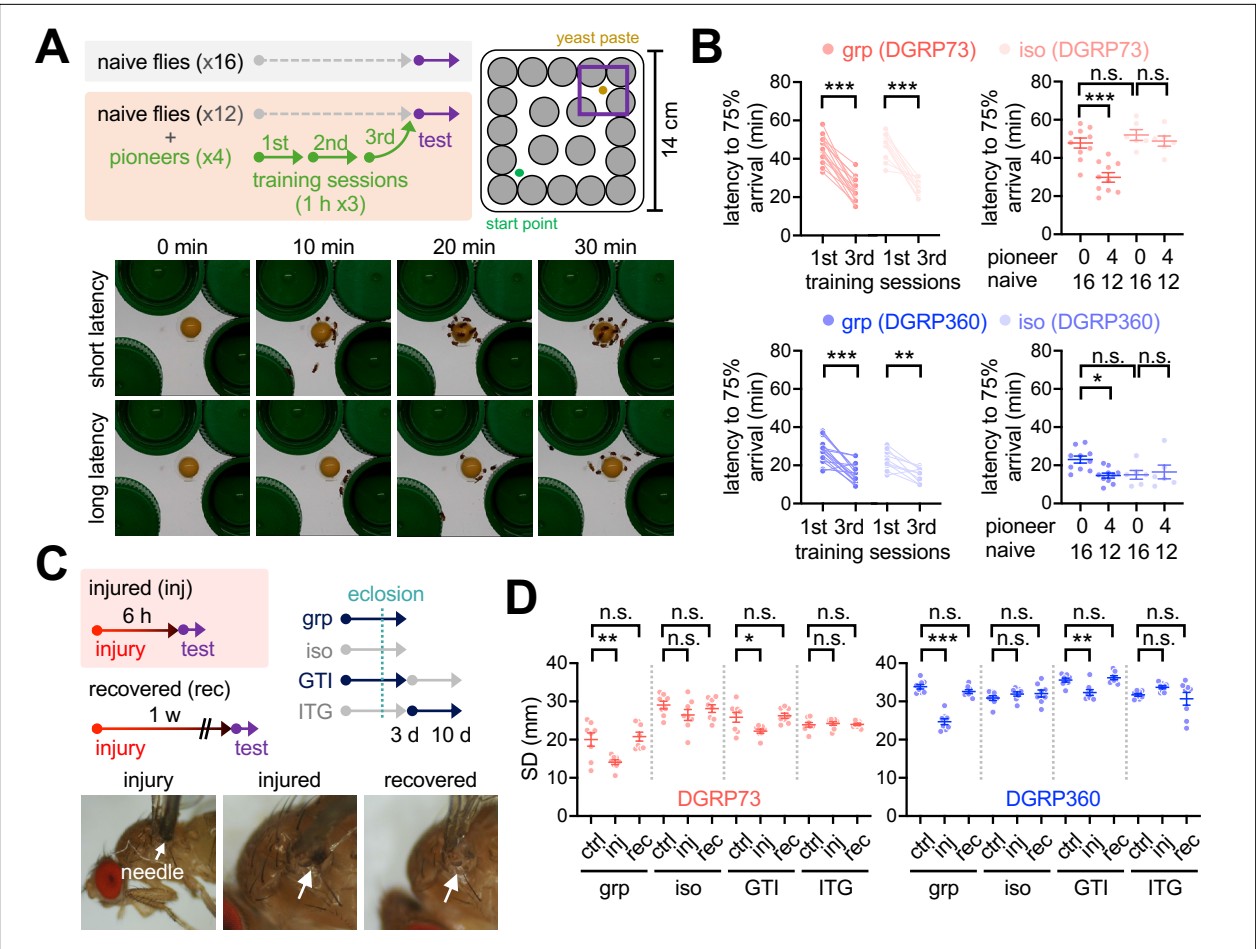

**Figure 4.** Early-life social experience confers beneficial effects on social foraging in adult *Drosophila*. (**A**) Experimental scheme for assessing social interactions in a maze assay. Representative images were shown for a group of *Drosophila* strains with high (short latency) or low social foraging (long latency). (**B**) Pioneer groups of flies from either grouped (grp) or isolated cultures (iso) were effectively trained in the maze assay, but social isolation of both short- (red, DGRP73) and long-SD flies (blue, DGRP360) blunted the pioneer effects on food-seeking behaviors in a group of naive flies. Two-way repeated measures ANOVA detected significant effects of training (p<0.0001 for DGRP73 and DGRP360) but not social isolation on latency during the training session of pioneer groups. Ordinary two-way ANOVA also detected significant interaction effects of pioneer and social isolation on latency during the maze test (p=0.0131 for DGRP73; p=0.0310 for DGRP360). Data represent means ± SEM (n = 6–16). n.s., not significant; *p<0.05, **p<0.01, ***p<0.001, as determined by Sidak's multiple comparisons test (training session) or Tukey's multiple comparisons test (maze test). (**C**) Experimental scheme for assessing injury-induced social network behavior (SNB) plasticity. GTI, grouped-to-isolated culture transition; ITG, isolated-to-grouped culture transition. (**D**) Physical injury induced clustering behaviors in group-cultured but not developmentally isolated flies. Data represent means ± SEM (n = 8). n.s., not significant; *p<0.05, **p<0.01, ***p<0.001, as determined by one-way ANOVA with Tukey's multiple comparisons test.

The online version of this article includes the following source data and figure supplement(s) for figure 4:

**Source data 1.** Quantitative analysis of adult SNB and social plasticity.

**Figure supplement 1.** Social isolation blunts pioneer effects on food-seeking behaviors in a group of naive flies.

**Figure supplement 2.** *Rutabaga*-dependent working memory is dispensable for social memory.

**Figure supplement 3.** Early-life social experience is necessary for social behavior plasticity in male flies.

**Figure supplement 4.** Mechanical injury reduces walking speed and centroid velocity only in group-cultured long-social distance (SD) flies.

**Figure supplement 5.** Loss of *norpA* function blocks injury-induced clustering in group-cultured male flies.

grouping of developmentally isolated flies was insufficient to support the social plasticity (*Figure 4D*, *Figure 4—figure supplement 3A*, *Figure 4—source data 1*). Finally, the *rut*-dependent memory pathway seems dispensable for developmental 'social memory' since *rut* mutants displayed injury-induced social plasticity comparable to control flies (*Figure 4—figure supplement 2B*, *Figure 4—source data 1*). These results suggest that individual *Drosophila* strains differentially display social

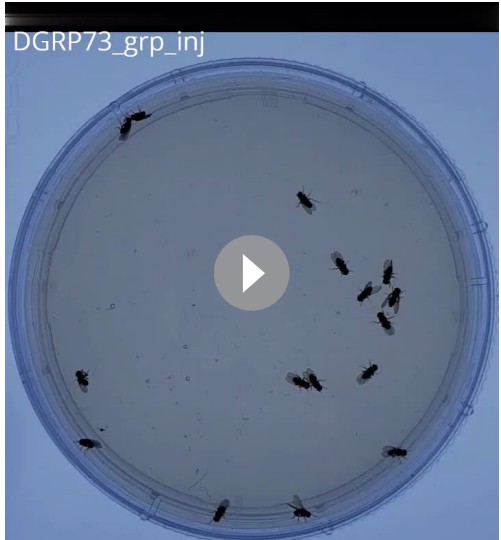

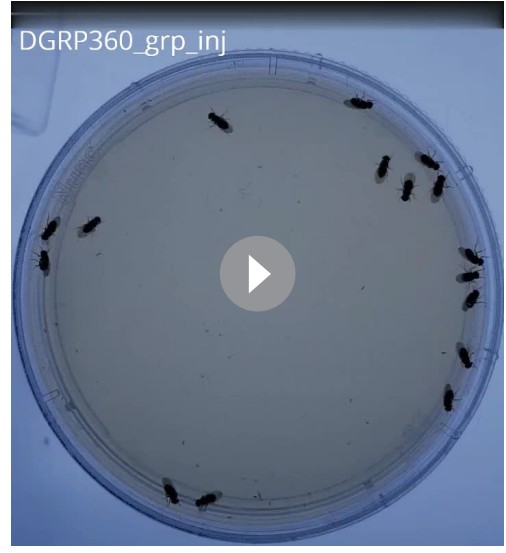

**Video 5.** Social network behavior (SNB) in DGRP73 (grp+inj).
https://elifesciences.org/articles/103973/figures#video5

**Video 6.** Social network behavior (SNB) in DGRP360 (grp+inj).
https://elifesciences.org/articles/103973/figures#video6

traits in adults; however, the potency of adaptive social plasticity is acquired through their early-life experience of social interactions.

## Unraveling the genetic basis of SNB and its plasticity

How is early-life social experience encoded in individual larvae to persist throughout development? To define the molecular signatures of social interaction phenotypes and their plasticity, we profiled differentially expressed genes (DEGs) in adult fly heads among distinct contexts of genetic and environmental sociality. The transcriptome analyses revealed significantly upregulated genes in the short- and long-SD lines (n genes = 191 and 199, respectively), as well as in grouped and socially isolated flies (n genes = 159 and 755, respectively) (*Figure 5A and B*, *Figure 5—source data 1*, *Figure 5—source data 2*). Genes upregulated in socially isolated DGRP lines overlapped substantially with those identified in previous studies using independent wild-type strains (*Li et al., 2021*; *Wang et al., 2008*;

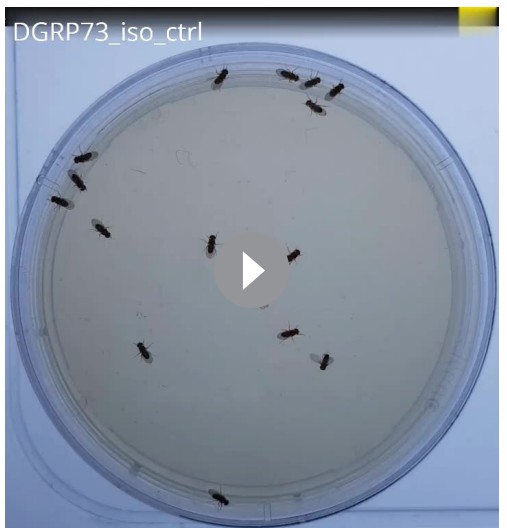

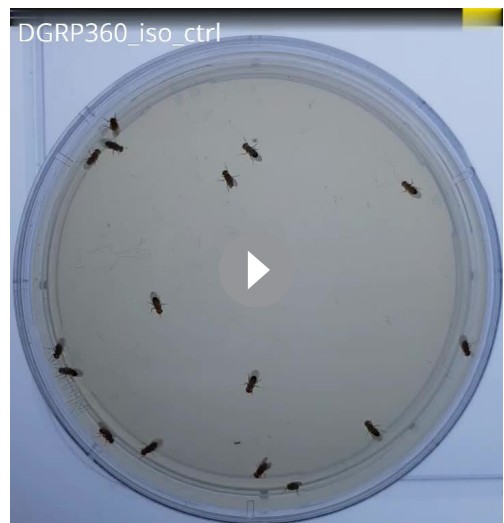

**Video 7.** Social network behavior (SNB) in DGRP73 (iso+ctrl).
https://elifesciences.org/articles/103973/figures#video7

**Video 8.** Social network behavior (SNB) in DGRP360 (iso+ctrl).
https://elifesciences.org/articles/103973/figures#video8

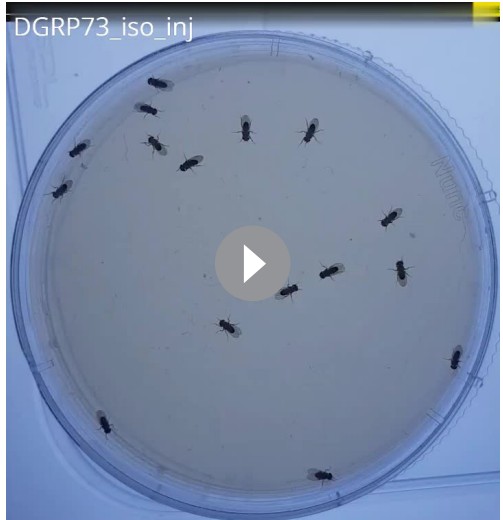

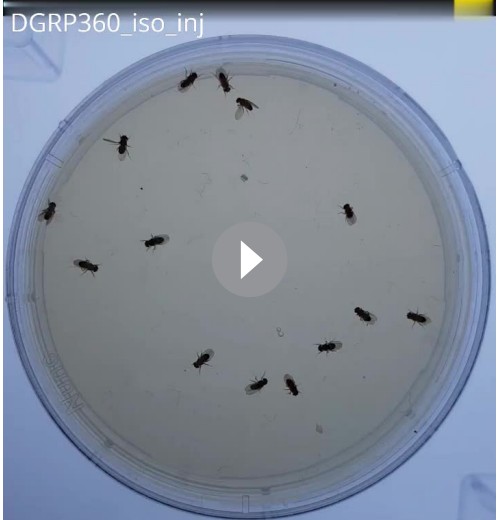

**Video 9.** Social network behavior (SNB) in DGRP73 (iso+inj).
https://elifesciences.org/articles/103973/figures#video9

**Video 10.** Social network behavior (SNB) in DGRP360 (iso+inj).
https://elifesciences.org/articles/103973/figures#video10

*Agrawal et al., 2020*; *Figure 5—figure supplement 1A*, *Figure 5—source data 3*), whereas upregulated genes in short- or long-SD lines barely overlapped with DEGs between grouped and socially isolated flies (*Figure 5—figure supplement 1B*, *Figure 5—source data 3*). Gene Ontology (GO) analyses showed no significant enrichment of specific GO terms in DEGs between the short- and long-SD lines. We thus concluded that diverse genetic pathways shape baseline SD phenotypes, as suggested previously (*Schneider et al., 2012*). It is also consistent with the lack of the phylogenetic correlation between *Drosophila* species and their social network phenotypes (*Jezovit et al., 2020*). By contrast, select metabolic pathways were upregulated in both types of DGRP lines by social isolation (*Figure 5C and D*, *Figure 5—source data 2*; *Figure 5—source data 4*). These observations were consistent with previous findings that chronic social isolation acts as a hunger cue to suppress sleep, induce feeding, and alter metabolic gene expression, including those involved in lipid metabolism (*Li et al., 2021*; *Liu et al., 2018*). Considering that the number of commonly upregulated genes in group-cultured flies was very limited, we speculate that *Drosophila* has evolved a genetic reprogram where social isolation elevates metabolic gene expression to adaptively induce a metabolic shift for energy storage and fitness.

Interestingly, the phenotypic alignment of DGRP lines revealed significant correlations of their social interaction behaviors to food intake, starvation-induced activity, and aggression, among others (*Garlapow et al., 2015*; *Shorter et al., 2015*; *Chi et al., 2021*; *Figure 5E*, *Figure 5—source data 5*). We validated that the long-SD lines showed more lunges than the short-SD lines, indicative of high aggression behaviors (*Figure 5—figure supplement 2*, *Figure 5—source data 5*, *Videos 11 and 12*). The association of multiple behaviors raised the possibility of their overlapping evolution of regulatory genes and mechanisms. The expression heatmaps of relevant gene categories visualized a subset of genes that indeed displayed social experience-dependent expression across the DGRP lines analyzed (*Figure 5F*, *Figure 5—source data 1*). These included *Drosulfakinin* (*Dsk*), a neuropeptide implicated in aggression, food intake, satiety, and sexual behaviors (*Agrawal et al., 2020*; *Wu et al., 2019*; *Wu et al., 2020*; *Guo et al., 2021*; *Wang et al., 2022*). The mammalian *Dsk* homolog cholecystokinin (CCK) has also been shown to play a similar role in relevant physiology, suggesting the possible conservation of *Dsk* function (*Nichols et al., 1988*; *Staljanssens et al., 2011*; *Nässel and Wu, 2022*). In fact, *Dsk* was the only overlapping gene that was downregulated upon social isolation across independent studies (*Li et al., 2021*; *Wang et al., 2008*; *Agrawal et al., 2020*; *Figure 5—figure supplement 1A*, *Figure 5—source data 3*). We also found that social isolation of the DGRP lines significantly downregulated the expression of *Dsk* and the two CCK-like receptors (i.e., *CCKLR-17D1* and *CCKLR-17D3*), although *Dsk* receptor gene expression showed less than twofold

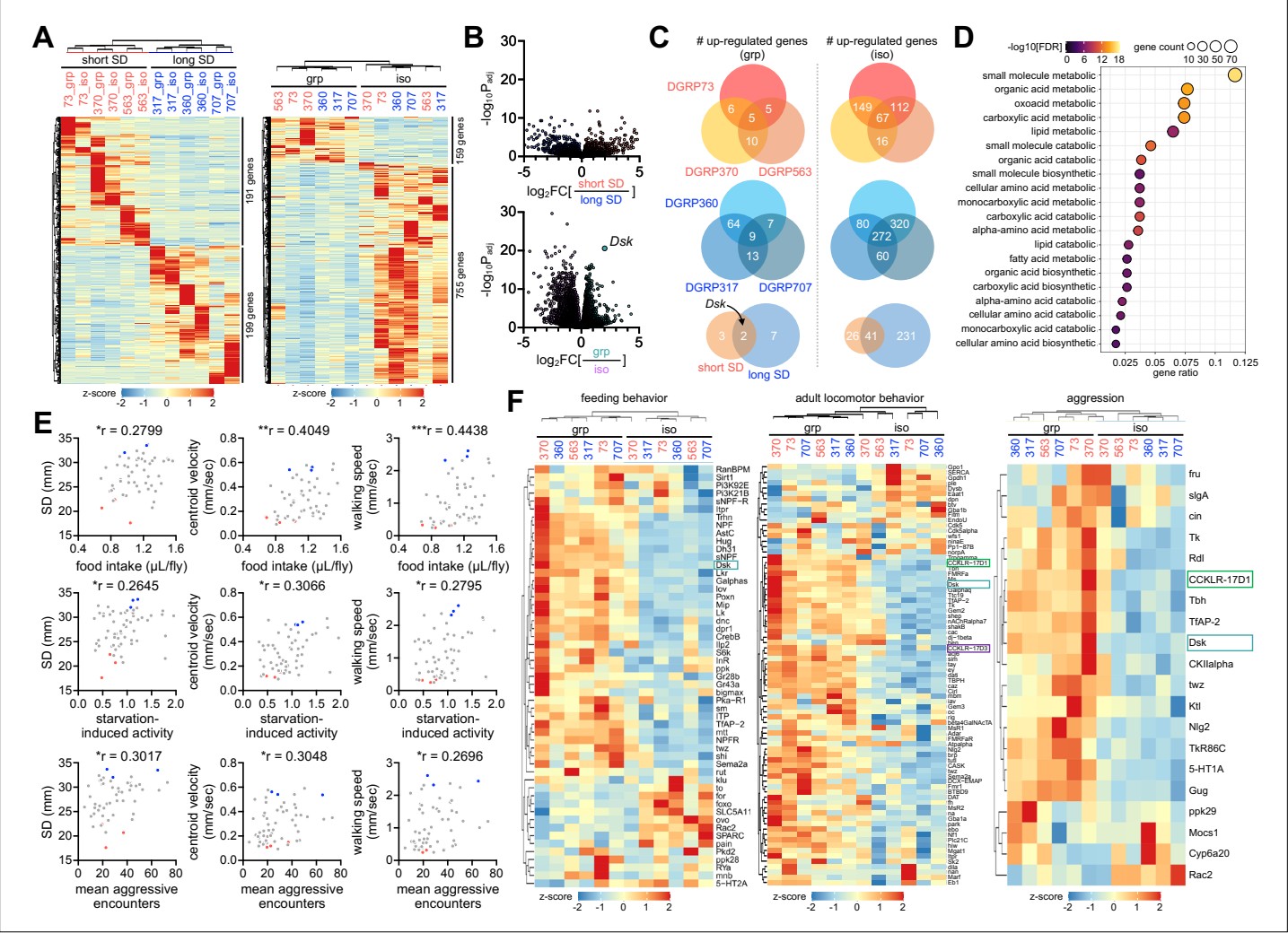

**Figure 5.** Social experience shapes gene expression profiles in *Drosophila* heads. (**A**) Heatmaps for differentially expressed genes (DEGs, more than twofold difference with adjusted p<0.05) in short- vs. long-social distance (SD) lines (left); in grouped (grp) vs. isolated (iso) condition (right). Fly heads were harvested from individual *Drosophila* Genetics Reference Panel (DGRP) lines in grouped or isolated cultures and their gene expression profiles were analyzed by RNA sequencing. Averaged counts per million were converted to z-score for visualization. (**B**) Volcano plots for differentially expressed genes (DEGs) in short- vs. long-SD lines (top); in grp vs. iso flies (bottom). Social interactions evidently upregulated the neuropeptide *Drosulfakinin* (*Dsk*) expression. (**C**) Overlapping DEGs in grp vs. iso conditions across DGRP lines. *Dsk* was identified as a commonly upregulated gene by social interactions in short- (top, red) and long-SD lines (middle, blue). (**D**) Gene Ontology analysis reveals upregulation of select metabolic pathways upon social isolation. False discovery rate (FDR) < 0.05, as determined by Fisher's exact test (grp vs. iso). (**E**) Significant phenotypic correlation of SNB to food intake, starvation-induced activity, and mean aggressive encounters among DGRP lines. Raw data for food intake (n = 52 DGRP lines), starvation-induced activity (n = 60 DGRP lines), and aggression (n = 57 DGRP lines) were obtained from previous studies (*Garlapow et al., 2015*; *Shorter et al., 2015*; *Chi et al., 2021*) and then aligned to SD, centroid velocity, and walking speed in the corresponding DGRP lines that were measured by our SNB analyses. *p<0.05, **p<0.01, ***p<0.001, as determined by Spearman correlation analysis. (**F**) Expression heatmap for genes implicated in feeding, adult locomotor behavior, and aggression. Downregulation of *Dsk* and its two receptors (CCKLR-17D1 and CCKLR-17D3) by social isolation was visualized in relevant gene categories.

The online version of this article includes the following source data and figure supplement(s) for figure 5:

**Source data 1.** Normalized gene expression in individual DGRP lines under grouped vs. isolated culture conditions.

**Source data 2.** DEG analyses between distinct social groups (short vs. long SD; or grouped vs. isolated).

**Source data 3.** Comparative analyses of social group-specific DEGs from independent studies.

**Source data 4.** DEG analyses among individual DGRP lines.

**Source data 5.** Correlation of SNB to food intake, starvation-induced activity, and aggression among DGRP lines.

*Figure 5 continued on next page*

**Figure supplement 1.** Cross-study analyses of social context-dependent differentially expressed genes (DEGs) highlight *Dsk* as the most prominently upregulated gene under socially enriched conditions.

**Figure supplement 2.** Long-social distance (SD) flies display higher aggression than short-SD flies.

change (*Figure 5F*, *Figure 5—source data 2*). These observations prompted us to ask whether DSK signaling contributes to the plasticity of social behaviors through early-life experience.

## DSK neuron activity encodes early-life experience for SNB plasticity

Immunostaining of whole-mount brains identified three groups of DSK-expressing neurons with distinct neuroanatomical morphology (i.e., MP1a, MP1b, and MP3) (*Wu et al., 2019*; *Wu et al., 2020*; *Figure 6A*). The long-SD lines displayed relatively high DSK signals in the cell bodies compared to the short-SD lines (*Figure 6B*, *Figure 6—source data 1*). However, DSK levels in the neural projections were comparable between the two groups, raising the possibility that axonal transport or processing of the neuropeptide was limiting under the group-culture condition. Social isolation lowered DSK levels irrespective of the SD phenotype (*Figure 6A and B*, *Figure 6—source data 1*), consistent with our DEG analysis above. The social experience effects on DSK levels were most evident in the MP1a neuron projections. Live-brain imaging of the genetically encoded $Ca^{2+}$ indicator GCaMP showed that DSK neuron activity correlated with social experience and plasticity in a wild-type background. Social deprivation reduced relative $Ca^{2+}$ levels in DSK neurons, whereas mechanical injury generally elevated DSK neuron activity (*Figure 6C*, *Figure 6—source data 1*). The two conditions, however, acted independently on the GCaMP signals in DSK neurons since their interaction effects were not significantly detected. Nonetheless, MP1a neuron activity did not respond to injury when transgenic flies were socially deprived during development, which may contribute to the lack of social behavior plasticity upon isolation (*Figure 6C*, *Figure 6—figure supplement 1*, *Figure 6—source data 1*). These observations are unlikely due to transgenic *Dsk*-Gal4 activity per se given that social isolation did not comparably affect the Gal4-dependent expression of a dendritic marker transgene in the DSK neurons (*Figure 6—figure supplement 1*, *Figure 6—source data 1*). We also confirmed that GCaMP levels in other neuropeptide-expressing neurons (i.e., pigment-dispersing factor) were insensitive to social isolation or mechanical injury (*Figure 6—figure supplement 2*, *Figure 6—source data 1*).

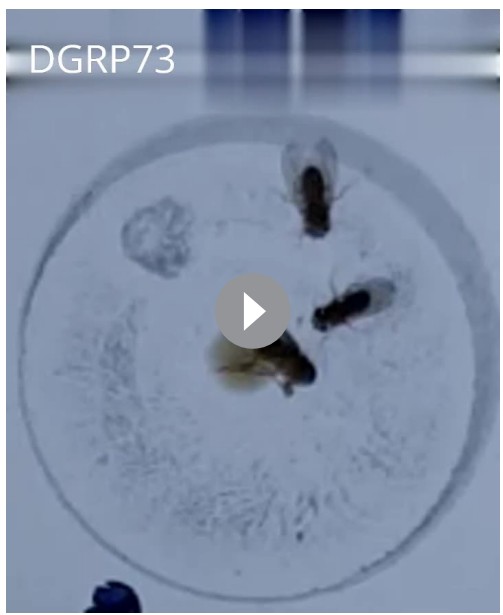

**Video 11.** Aggression behaviors in DGRP73.
https://elifesciences.org/articles/103973/figures#video11

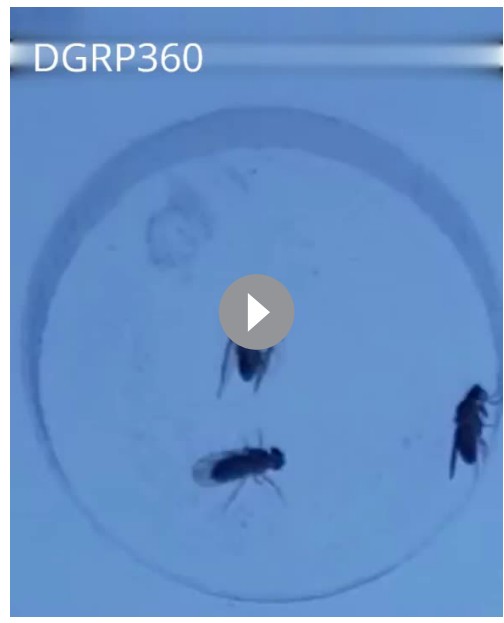

**Video 12.** Aggression behaviors in DGRP360.
https://elifesciences.org/articles/103973/figures#video12

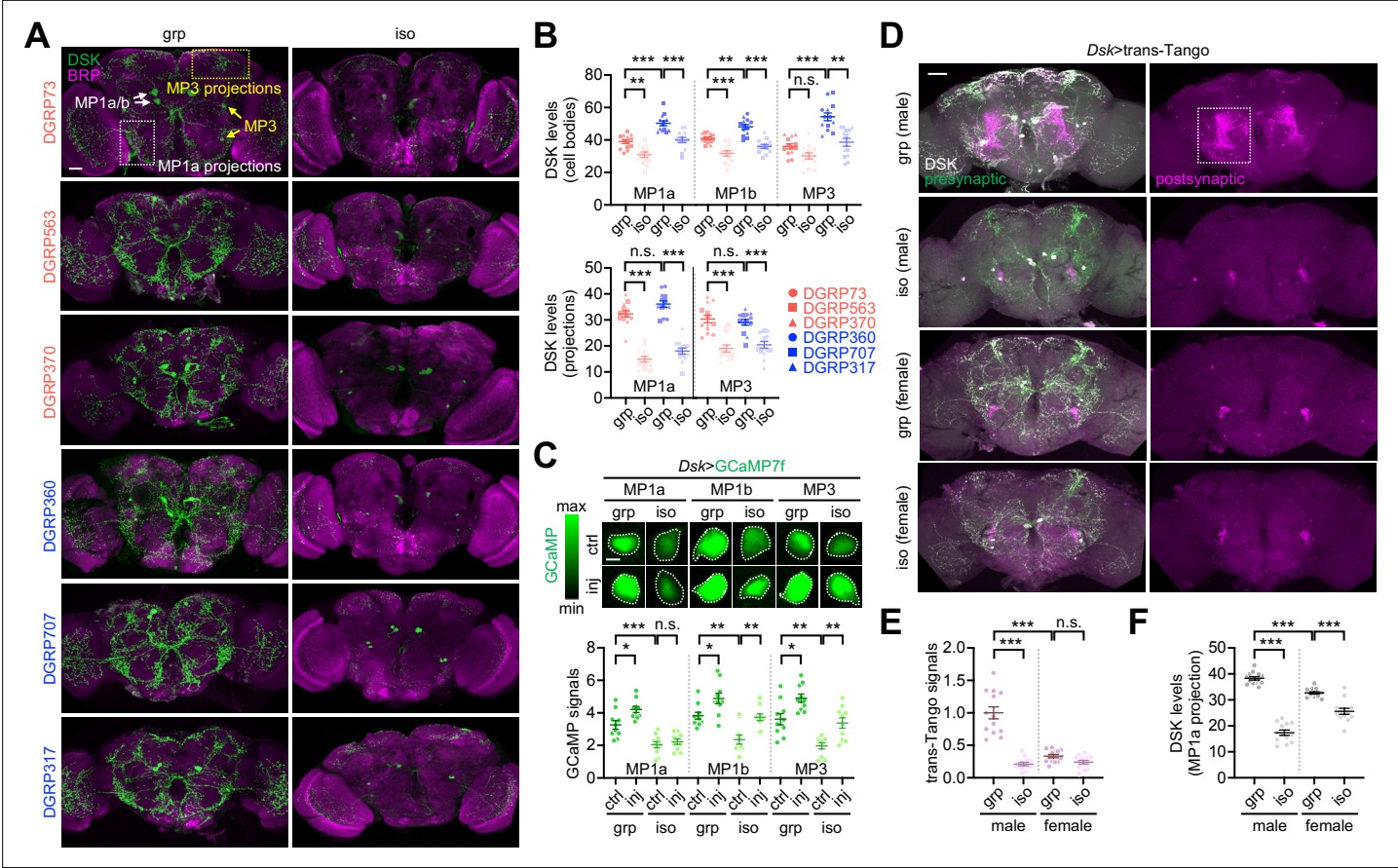

**Figure 6.** Drosulfakinin (*Dsk*) neuron activity encodes social experience. (**A, B**) Social experience elevates DSK levels in the *Drosophila* brains. Whole-mount brains from each *Drosophila* Genetics Reference Panel (DGRP) line in grouped (grp) or socially isolated cultures (iso) were co-immunostained with anti-DSK (green) and anti-BRUCHPILOT antibodies (BRP, a synaptic protein; magenta). The fluorescent DSK signals from confocal images were quantified using ImageJ. Ordinary two-way ANOVA or aligned ranks transformation ANOVA detected significant effects of social isolation on DSK levels in MP1a/MP1b/MP3 cell bodies and their projections ($p<0.0001$). The social distance (SD) trait effects (i.e., short vs. long SD) were more evident on DSK levels in cell bodies ($p<0.0001$). Data represent means ± SEM (n = 13). n.s., not significant; **$p<0.01$, ***$p<0.001$, as determined by Tukey's multiple comparisons test (MP1a/MP1b cell bodies and MP1a projections) or Wilcoxon rank sum test (MP3 cell bodies and projections). Scale bar = 40 µm. (**C**) Social experience and physical injury elevate DSK neuron activity. Social isolation masked an injury-induced $Ca^{2+}$ increase in the MP1a neurons among other DSK neurons as assessed by the genetically encoded $Ca^{2+}$ sensor GCaMP in live-brain imaging. Two-way ANOVA detected significant effects of social isolation ($p<0.0001$) and injury (inj, $p<0.01$) on GCaMP levels in DSK neurons. Data represent means ± SEM (n = 10). n.s., not significant; *$p<0.05$, **$p<0.01$, ***$p<0.001$, as determined by Tukey's multiple comparisons test. Scale bar = 5 µm. (**D–F**) Social experience strengthens postsynaptic signaling of DSK neurons only in males. Whole-mount brains were dissected from group-cultured (grp) or isolated (iso) flies and immunostained with anti-DSK antibody (white). Postsynaptic partner of DSK neurons was visualized by the heterologous ligand-receptor signaling embedded in the transsynaptic mapping transgene (*Dsk*>trans-Tango, magenta) while DSK neurons were further labeled by the presynaptic marker of the trans-Tango (green). The fluorescent trans-Tango signals and anti-DSK staining intensities from confocal images were quantified using ImageJ. Aligned ranks transformation ANOVA or ordinary 2-way ANOVA detected significant interaction effects of social isolation and gender on trans-Tango signals (**E**, $p<0.0001$) and DSK levels in MP1a projection (**F**, $p<0.0001$). Data represent means ± SEM (n = 13). n.s., not significant; ***$p<0.001$, as determined by Wilcoxon rank sum test (trans-Tango signals) or Tukey's multiple comparisons test (DSK levels). Scale bar = 40 µm.

The online version of this article includes the following source data and figure supplement(s) for figure 6:

**Source data 1.** Quantitative analysis of DSK neuron activities under distinct social contexts.

**Figure supplement 1.** Neither social experience nor physical injury affects transgenic DenMark expression in DSK neurons.

**Figure supplement 2.** Neither social experience nor physical injury affects circadian-clock neuron activity.

We used the transsynaptic mapping transgene trans-Tango to visualize the postsynaptic partner of DSK neurons via heterologous ligand-receptor signaling (*Talay et al., 2017*; *Sorkaç et al., 2023*). Socially enriched, but not socially isolated, male flies displayed strong postsynaptic signals of the trans-Tango-expressing DSK neurons in the lateral protocerebrum where transgenic signals of DSK

neuron axons (*Dsk*>SytGFP) were readily detectable (*Figure 6D and E*, *Figure 6—figure supplement 1B*, *Figure 6—source data 1*). Previous studies also demonstrated that MP1a neuron projections are specifically enriched in this brain region (*Wu et al., 2019*; *Wu et al., 2020*). DSK neurons have gender-specific presynaptic partners, which may mediate distinct signaling for sexual behaviors (*Wu et al., 2019*; *Wu et al., 2020*; *Wang et al., 2022*). To our surprise, socially enriched female brains did not show trans-Tango signals as evidently as male brains (*Figure 6D and E*, *Figure 6—source data 1*). Moreover, social deprivation downregulated DSK levels in the MP1a projections less potently in females than in males (*Figure 6D and F*, *Figure 6—source data 1*). These observations suggest sexual dimorphism in social behavior plasticity. Indeed, female flies from select DGRP lines showed baseline SD phenotypes consistent with their male counterparts, yet mechanical injury did not significantly affect female SD (*Figure 4—figure supplement 3B*, *Figure 4—source data 1*; also see *Figure 7G*, *Figure 7—source data 1*). Considering that DSK-expressing MP1 neurons originate from the larval brain (*Oikawa et al., 2023*), we hypothesized that early-life experience is developmentally encoded in DSK neuron activity via the male-specific neural pathway and adaptively expressed for social plasticity in adults. We further reason that low food intake upon social isolation may not directly implicate DSK expression or DSK neuron activity since DSK signaling has been shown to suppress feeding behavior as a satiety cue (*Wu et al., 2020*; *Guo et al., 2021*; *Söderberg et al., 2012*; *Williams et al., 2014*).

## Male-specific DSK-CCKLR-17D1 signaling mediates SNB plasticity

To determine whether DSK signaling actually contributes to social behavior plasticity, we first examined the loss-of-function effects of relevant genes. Genomic deletion of the *Dsk* locus (*Dsk*^attP^) or DSK depletion by RNA interference (*Dsk*>*Dsk*^RNAi^) abolished injury-induced clustering behaviors (*Figure 7A–C*, *Figure 7—figure supplement 1*, *Figure 7—source data 1*). Furthermore, genomic deletions of the *Dsk* receptor *CCKLR-17D1* (*CCKLR-17D1*^attP^ and *CCKLR-17D1*^Δ^) but not *CCKLR-17D3* (*CCKLR-17D3*^attP^ and *CCKLR-17D3*^Δ^) comparably masked injury-induced social interactions (*Figure 7B*, *Figure 7—figure supplement 2*, *Figure 7—source data 1*). Genetic effects of *Dsk* deletion and DSK depletion were not consistent on baseline SD in grouped cultures. These observations suggest that *Dsk* is not crucial for shaping the SD traits per se, but genetic backgrounds may substantially contribute to it. Nonetheless, genetic evidence from independent alleles and transgenic RNAi convincingly supports specific implication of the DSK-CCKLR-17D1 pathway in experience-dependent social plasticity.

We further validated if neuronal activity or synaptic transmission for DSK-CCLKR-17D1 signaling controls injury-induced SNB plasticity. To this end, a temperature-sensitive allele of *Drosophila* dynamin (*shibire*^ts^) (*Kitamoto, 2001*) was transgenically expressed in DSK neurons to block their synaptic transmission only at restrictive temperatures (*Figure 7D*). The conditional manipulation of larval DSK neurons was sufficient to suppress injury-induced clustering in group-cultured adults (*Figure 7E*, *Figure 7—source data 1*). CCKLR-17D1-expressing neurons displayed a gender-specific distribution in the adult brain when their dendrites and axons were visualized using specific transgenic markers (i.e., DenMark and SytGFP, respectively) (*Figure 7F*). In particular, male brains expressing the CCKLR-17D1 knock-in transgene showed more signals for cell bodies and dendrites around MP1a axon projections than females (*Wu et al., 2020*). This was consistent with the high levels of axonal DSK expression and trans-Tango signals in the same region of male brains (*Figure 6D–F*, *Figure 6—source data 1*). We hypothesized that transgenic activation of DSK-CCKLR-17D1 signaling genetically mimics social experience in developmentally isolated flies. Supporting this idea, transgenic excitation of CCKLR-17D1 neurons was sufficient to confer injury-induced social interactions to isolated male flies (*Figure 7G*, *Figure 7—source data 1*). The same transgenic manipulation of CCKLR-17D1 neurons did not induce social plasticity in females, irrespective of their early-life experience. On the other hand, blocking synaptic transmission in CCKLR-17D1 neurons suppressed injury-induced clustering in group-cultured male flies (*Figure 7—figure supplement 3*, *Figure 7—source data 1*). These findings convincingly provide a neuroanatomical basis for early-life social memory and male-specific social plasticity.

## Discussion

Our study demonstrated that conspecific individuals show a wide range of preferences for social distancing, and it has likely co-evolved with their inferior traits as a compensatory mechanism

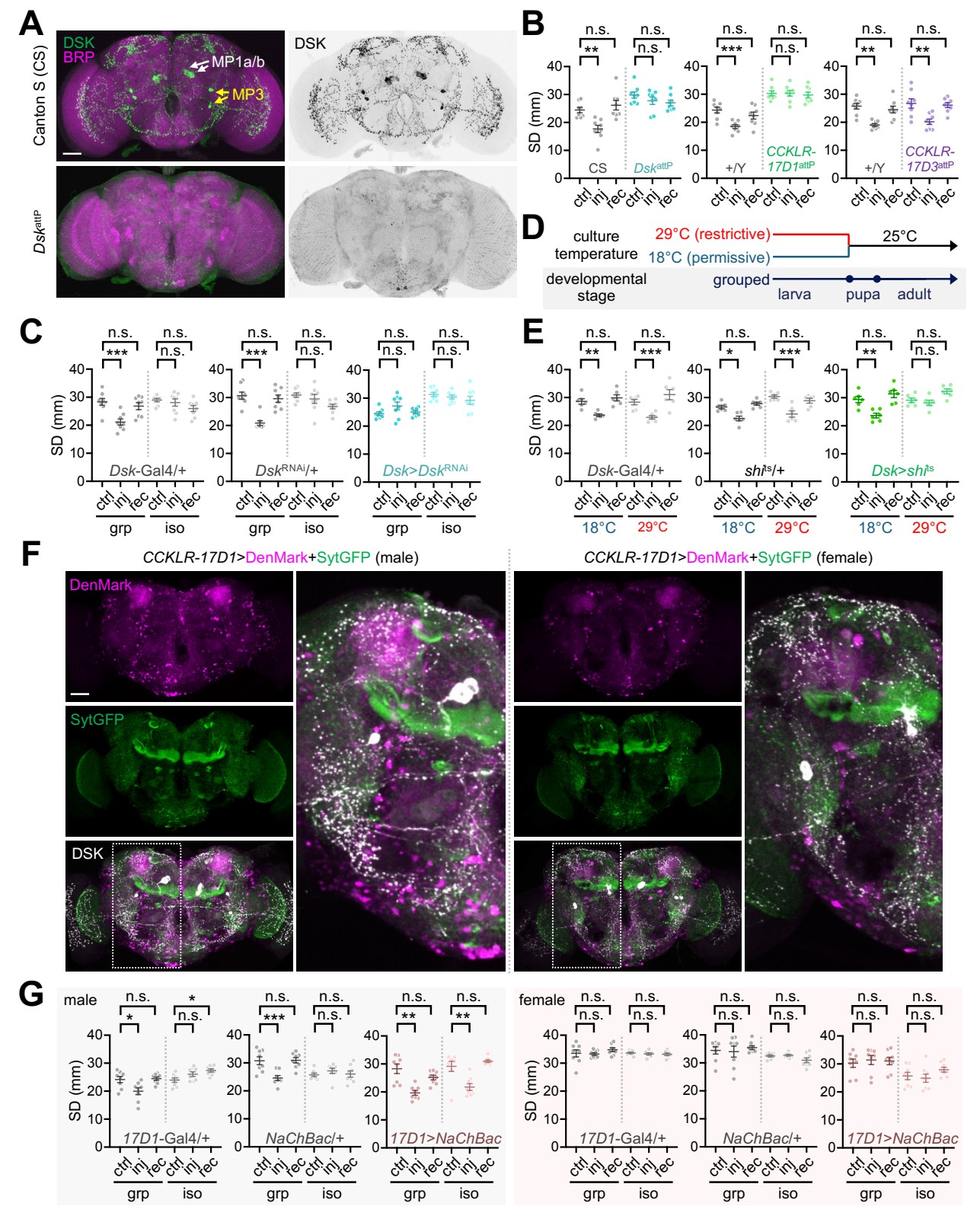

**Figure 7.** Genetic manipulations of Drosulfakinin (DSK) signaling imitate social experience. (**A**) *Dsk*^attP mutant brain expresses barely detectable DSK peptides. Whole-mount brains from Canton S (CS, a wild-type control) and *Dsk*^attP mutant flies were co-immunostained with anti-DSK (green) and anti-BRUCHPILOT antibodies (BRP, a synaptic protein; magenta). Scale bar = 40 μm. (**B, C**) Genetic silencing of DSK-CCKLR-17D1 signaling by genomic deletions (*Dsk*^attP or *CCKLR-17D1*^attP) or DSK depletion (Dsk >*Dsk*^RNAi) blocks injury-induced clustering behaviors. Data represent means ± SEM (n = 8).

*Figure 7 continued on next page*

*Figure 7 continued*

n.s., not significant; **p<0.01, ***p<0.001, as determined by two-way ANOVA with Tukey's multiple comparisons test. (**D, E**) Conditional blockade of DSK neuron transmission at larval stage is sufficient to blunt social experience-dependent plasticity of social network behavior (SNB). Transgenic crosses for DSK-specific expression of the temperature-sensitive *shibire* allele (Dsk >*shi*ts) were kept at either restrictive (29°C) or permissive temperature (18°C) for synaptic transmission until the end of larval stage. Two-way ANOVA detected significant interaction effects of injury (inj) and temperature on social distance (SD) in Dsk>*shi*ts flies (p=0.0169) but not in heterozygous controls (p=0.7549 for *Dsk*-Gal4/+; p=0.2030 for *shi*ts/+). Data represent means ± SEM (n = 6). n.s., not significant; *p<0.05, **p<0.01, ***p<0.001, as determined by Tukey's multiple comparisons test. (**F**) CCKLR-17D1 neurons display sexually dimorphic dendrites around MP1 projections. Transgenic male or female brains (*CCKLR-17D1*>DenMark + SytGFP) were immunostained with anti-DSK antibody to visualize DSK neuron projections (white) along with dendrites (DenMark, magenta) and axons (SytGFP, green) of neurons expressing CCKLR-17D1-Gal4 knock-in. Scale bar = 40 μm. (**G**) Transgenic excitation of CCKLR-17D1 neurons confers injury-induced plasticity of SNB in socially isolated males but not females. Two-way ANOVA detected significant interaction effects of injury (inj) and social isolation on SD in male heterozygous controls (p=0.0040 for 17D1-Gal4/+; p=0.0007 for NaChBac/+) but not in all the other genotypes. Data represent means ± SEM (n = 8). n.s., not significant; *p<0.05, **p<0.01, as determined by Tukey's multiple comparisons test.

The online version of this article includes the following source data and figure supplement(s) for figure 7:

**Source data 1.** Quantitative analysis of SNB plasticity in genetic and transgenic *Drosophila* models for DSK-DSK receptor signaling.

**Figure supplement 1.** Transgenic *Dsk* RNAi effectively depletes DSK peptides in the adult brain.

**Figure supplement 2.** Genomic deletions of *CCKLR-17D1* but not *CCKLR-17D3* suppress injury-induced clustering in group-cultured male flies.

**Figure supplement 3.** Transgenic silencing of CCKLR-17D1 neurons suppresses injury-induced clustering in group-cultured male flies.

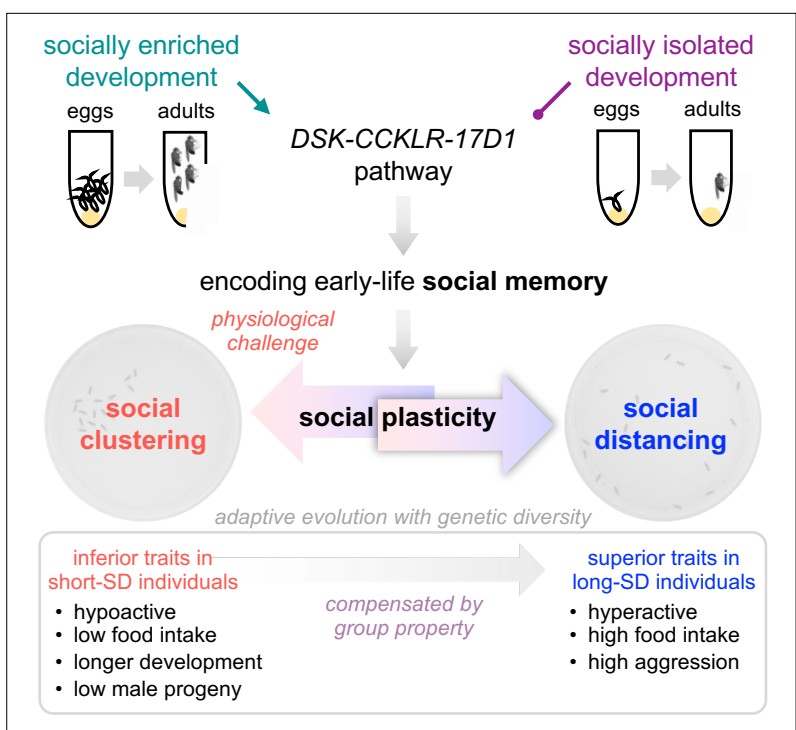

**Figure 8.** A working model for early-life social memory and the experience-dependent plasticity of social network behavior (SNB). Social distancing is an inheritable trait in *Drosophila*. Diverse genetic pathways contribute to active social preferences while the clustering property may have evolved to compensate for inferior traits in individuals and confer their group fitness. Nonetheless, *Drosophila* can tune their social distance depending on physiological states (e.g., mechanical injury) and this feature of SNB is defined as 'social plasticity'. In fact, the social plasticity requires early-life social experience or 'social memory' during development. Group culturing elevates DSK expression and DSK neuron activity to reinforce its postsynaptic signals likely to the cognate receptor CCKLR-17D1. The activation of DSK-CCKLR-17D1 pathway thus encodes early-life social experience in developing brains to support social plasticity in adults.

(*Figure 8*). Physiological challenges (e.g., mechanical injury) may adaptively modify the clustering property of a given group. However, the plasticity of group behaviors requires social experience during development. We propose that a subset of the *Drosophila* brain neurons expressing the neuropeptide DSK serves as a neural substrate for social memory, as supported by the intimate coupling of early-life social experience and SNB plasticity to DSK expression, DSK neuron activity, and postsynaptic signaling.

Distinct social behaviors in *Drosophila* (i.e., aggression and mating) have been commonly mapped to specific pairs of DSK neurons (*Wu et al., 2019*; *Wu et al., 2020*; *Wang et al., 2022*). Presynaptic partners of DSK neurons are sexually dimorphic, while their postsynaptic effects are differentially mediated via the two DSK receptor pathways (i.e., CCKLR-17D1 for aggression and SNB plasticity; CCKLR-17D3 for sexual behaviors). Intriguingly, this neural architecture for balancing aggression and mating behaviors has been proposed to be analogously conserved between flies and mammals (*Anderson, 2016*). Furthermore, DSK neuron activity correlates with social dominance (i.e., winner effects from aggression) (*Wu et al., 2020*) and group housing (*Wu et al., 2019*), but not with mating status (*Wang et al., 2022*). Accordingly, the DSK-CCKLR-17D1 pathway meets the necessary criteria for male-specific SNB plasticity.

The phenotypic correlates of aggression and SNB in natural populations (i.e., DGRP lines) are consistent with their common neural locus. However, these findings do not necessarily imply that DSK neurons control the two social behaviors in a similar manner. For instance, DSK excitation promotes aggression in both males and females (*Wu et al., 2020*), whereas DSK signaling is unlikely to trigger grouping behaviors per se. What remains to be clarified is how DSK signaling gates SNB plasticity and why this process is missing in female flies. Considering that social hierarchy is established in male fights only (*Nilsen et al., 2004*; *Simon and Heberlein, 2020*), we speculate that male-specific DSK pathways may include extra circuit modalities for contextual processing of social environments and adaptive social structures. The development of neuroanatomical differences between male and female brains may coincide with the acquisition of gender-specified demands for innate behaviors and physiology during evolution (e.g., a reproductive advantage of male clustering under physiologically challenging conditions).

The mammalian DSK homolog CCK shows sexual dimorphism in brain expression and mating behavior response (*Bloch et al., 1987*; *Bloch et al., 1988*; *Micevych et al., 1988*). Moreover, CCK activation is implicated in aggression and exploratory behaviors (*Raud et al., 2005*; *Li et al., 2007*). It would thus be interesting to determine whether DSK/CCK signaling indeed represents an ancestral mechanism for social memory, sexually dimorphic social behaviors, and their plasticity.

## Materials and methods

### Fly stocks

Flies were raised on standard cornmeal-yeast-agar food at 25°C and 40–50% humidity under 12 hr light:12 hr dark cycles. Behavioral experiments were primarily conducted between Zeitgeber time (ZT) 4 and 8 (lights-on at ZT0; lights-off at ZT12). DGRP lines, Canton-S (BL64349), $rut^1$ (BL9404), $rut^{2080}$ (BL9405), $Dsk^{attP}$ (BL84497), $Dsk^{2A-Gal4}$ (BL84630), UAS-$Dsk^{RNAi}$ (BL25869), $CCKLR$-$17D1^{attP}$ (BL84462), $CCKLR$-$17D1^{2A-Gal4}$ (BL84605), $CCKLR$-$17D3^{attP}$ (BL84463), $Orco^1$ (BL23129), UAS-myrGFP.QUAS-mtdTomato-3xHA; trans-Tango (BL77124), 20XUAS-IVS-jGCaMP7f (BL80906), and UAS-DenMark, UAS-syt.eGFP (BL33065) were obtained from Bloomington *Drosophila* Stock Center. $CCKLR$-$17D1^{\Delta1}$ (119026), $CCKLR$-$17D1^{\Delta2}$ (119027), $CCKLR$-$17D3^{\Delta1}$ (119029), $CCKLR$-$17D3^{\Delta2}$ (119030), $iav^1$ (101174), $norpA^7$ (108362), and $Poxn^{68}$ (119155) were obtained from Kyoto *Drosophila* Stock Center. UAS-$shi^{ts}$, UAS-NaChBac, and UAS-TNT have been described previously (*Kitamoto, 2001*; *Nitabach et al., 2006*).

### SNB analysis

A Petri dish (5.5 cm [d] × 1.5 cm [h]) was filled with 2% agar media (1.3 cm [h]) to prepare a circular arena for SNB analysis. This setup minimizes side-wall walking or z-stacking of individual flies that interferes with tracing group behaviors over time (*Simon and Dickinson, 2010*). Three-to-five-day-old flies from standard cultures (n = 16) were briefly cold-anesthetized (<15 s) and then transferred to the circular arena. Each arena was video-recorded for 10 min using a cellular phone (Samsung Galaxy

Note 8 or Samsung Galaxy Note 20). Time-series coordinates of each fly's position in the arena were extracted from raw video data using an in-house Python code (https://github.com/taejoonlab/tracking-fly, copy archived at *Jeong et al., 2021a*). SD was calculated from the position coordinates of individual flies at a given time and averaged over time. The average walking speed of individual flies and the interquartile ranges of the positional centroid in a given group of flies were also calculated over time (https://github.com/KJKwon/2023_FlyBehavior, copy archived at *Jeong et al., 2023*). The centroid velocity was determined from the fourth quarter of video-recording data, given more evident clustering phenotypes at the later period. Single-fly recordings in the circular arena were analyzed using a MATLAB-based fly_tracker code (https://github.com/jstaf/fly_tracker, copy archived at *Stafford, 2016*).

## Fly manipulations

*Drosophila* eggs were collected on a plate filled with grape juice media (https://cshprotocols.cshlp.org/content/2007/9/pdb.rec11113). Each egg was gently transferred to an isolation chamber containing 300 ul of cornmeal-yeast-agar food for social isolation. The chamber was sealed using parafilm with a tiny hole for air circulation and kept at 25°C before relevant experiments. To generate grouped-to-isolated (GTI) flies, 3-day-old flies from standard culture vials were individually transferred to each isolation chamber and further incubated for a week. To generate isolated-to-grouped (ITG) flies, 3-day-old flies eclosed from isolated eggs were collected into a standard food vial for group-culturing (~20 isolated flies per vial) and further incubated for a week. To assess SNB in isolated or GTI flies, each fly was reared in the isolation chamber before collectively transferring into the SNB arena. Flies were briefly cold-anesthetized during GTI/ITG transitions or before transferring to the SNB arena. For physical injury, the mesothoracic segment of 3-day-old flies was pierced (<1 mm depth) using a sterilized needle.

## Larval behavior and developmental analyses

For clustering assay, third-instar larvae were obtained from grouped-egg cultures (50 eggs per culture). A group of the third-instar larvae (n = 20) were then loaded onto a cylinder vial (2.3 cm [d] × 9.5 cm [h]) containing standard cornmeal-yeast-agar food. The food vial was divided into four sectors. The maximum number of clustering larvae per sector was scored from each vial, and the percentage of clusters with given larvae numbers was calculated from 30 vials. For digging assay, a 3D arena (2 cm [w] × 0.5 cm [d] × 4 cm [h]) was filled up to 2 cm with standard cornmeal-yeast-agar food. Either a group of third-instar larvae (n = 20) from standard culture vials or an isolated third-instar larva from single-egg cultures (one egg per culture) was transferred to the digging-assay arena and then allowed to explore it for 12 hr before measuring digging depth from the surface (*Dombrovski et al., 2017*). For food intake assay, a wider 3D arena (8 cm [w] × 0.5 cm [d] × 4 cm [h]) was filled up to 2 cm with standard cornmeal-yeast-agar media containing 1% brilliant blue FCF (JUNSEI, 64350-0410). A group of third-instar larvae (n = 20) from standard culture vials or an isolated third-instar larva from single-egg cultures was transferred to the food intake arena and then allowed to explore it for 12 hr. Each larva was gently homogenized in 50 ul of distilled water, and the absorbance of individual larval extracts was measured at 627 nm using a microplate reader (Tecan, Infinite M200). Developmental time was measured by the first eclosion day in a grouped-egg culture (50 eggs per culture) vs. a set of 81 single-egg cultures per experiment. The percentage of eclosed flies was scored from a grouped-egg culture (100 eggs per culture) vs. a set of 81 single-egg cultures per experiment, and the ratio of males to total flies was then calculated accordingly.

## Maze assay

Yeast paste was placed at the corner of a 14 cm × 14 cm transparent maze, and the maze was put on a white-light box to avoid any phototatic effects. A group of four flies was starved for 6 hr and then transferred to the maze by an aspirator. A training session was completed when three or more pioneer flies reached the food, and the trained flies were transferred to an empty vial containing water only. The pioneer training was repeated three times. For the maze test, a group of 16 flies (16 naive or 12 naive + 4 pioneer flies) was briefly cold-anesthetized (<15 s) and then placed at the opposite corner of the maze to the yeast paste. The 75% arrival time was recorded when 12 or more flies reached the food.

## Aggression assay

Quantitative assessment of aggression behaviors was performed as described previously with minor modifications (*Chen et al., 2002*). Three-to-five-day-old male flies were separated from females and starved for 6 hr. A pair of male flies with the same genotype were then transferred by gentle aspiration into a cylinder arena (1.4 cm [d] × 0.5 cm [h]), where a thin droplet of yeast paste with a decapitated female carcass was placed in the center. The arena was video-recorded for 10 min, and the number of lunges was counted manually.

## Courtship chaining behavior assay

Male–male courtship was quantified by the courtship chaining index as described previously with minor modifications (*Kitamoto, 2002*). A group of 3-to-5-day-old male flies (n = 16) from standard cultures were briefly cold-anesthetized (<15 s), transferred to the circular arena for SNB analysis, and then video-recorded for 10 min. Courtship chain was scored when three or more male flies were engaged in chaining with courtship behavior (*Kitamoto, 2002*). The chaining index was calculated by the percentage of courtship chaining duration over the 10 min recording.

## Transcriptome analysis

Three-to-five-day-old flies were harvested at ZT4-6. Total RNAs were extracted from 35 fly heads and purified using the PureLink RNA mini kit according to the manufacturer's instructions (Invitrogen). RNA quality was assessed by Bioanalyzer using the Agilent RNA 6000 pico kit (Agilent Technologies). RNA-seq libraries were constructed using the NEBNext Ultra Directional RNA Library Prep Kit for Illumina (New England Biolabs), together with NEBNext Poly(A) mRNA Magnetic Isolation Module (New England Biolabs) and subsequently sequenced by Illumina NovaSeq 6000 or Illumina NextSeq 500/550 (LabGenomics, Republic of Korea). RNA-seq reads were processed using trimmomatic (*Bolger et al., 2014*) (version 0.39) with the default option to remove bases with low-quality scores or from sequencing adapters. The trimmed reads were then mapped to the *Drosophila melanogaster* reference genome R6.46 using STAR (*Dobin et al., 2013*) (version 2.7.10b). DEGs were determined using DEseq2 (*Love et al., 2014*) (version 1.40.0; more than twofold change with adjusted p<0.05). Transcripts undetectable in more than half of the RNA-seq libraries were excluded from the analysis. Overrepresented GO terms were identified by Fisher's exact test (false discovery rate < 0.05) with PANTHER (*Thomas et al., 2022*) (version 17.0).

## Quantitative brain imaging

Whole-mount brain imaging was performed as described previously (*Jeong et al., 2021b*). The primary antibodies used in immunostaining included rabbit anti-DSK (Boster Bio, DZ41371; diluted at 0.5 ug/ml) and mouse anti-BRP antibodies (Developmental Studies Hybridoma Bank, nc82; diluted at 1:1,000). The GCaMP signals from live brains were recorded at room temperature using an FV1000 (Olympus) or A1 confocal microscope (Nikon). Fluorescence intensities from regions of interest in confocal images were quantified by background normalization [(S-B)/B] using ImageJ software.

## Statistical analysis

Statistical analyses were performed using GraphPad Prism or R (version 4.2.3). Shapiro–Wilk test was followed by F-test (two samples) or Brown–Forsythe test (multiple samples) to check normality (p<0.05) and equality of variances (p<0.05), respectively. For two-sample comparisons, parametric datasets with equal variance were analyzed by unpaired *t*-test. For multiple-sample comparisons, (1) parametric datasets with equal variance were analyzed by ordinary ANOVA with Tukey's multiple comparisons test; (2) parametric datasets with unequal variance were analyzed by Welch's ANOVA with Dunnett's T3 multiple comparisons test (one-way) or by aligned ranks transformation ANOVA with Wilcoxon rank sum test (two-way); and (3) nonparametric datasets with equal variance were analyzed by Kruskal–Wallis test with Dunn's multiple comparisons test (one-way) or by aligned ranks transformation ANOVA with Wilcoxon rank sum test (two-way). For comparisons between repeatedly measured samples, datasets were analyzed by paired *t*-test (two samples) or two-way repeated measures ANOVA with Sidak's multiple comparisons test (multiple samples). The significance of the correlation among SD, walking speed, centroid velocity, food intake, starvation-induced activity, and aggression in DGRP strains was determined by Spearman correlation analysis. Sample sizes and p

values obtained from individual statistical analyses were all summarized in each source data and indicated in the figure legends accordingly.

## Acknowledgements

We thank Bloomington *Drosophila* Stock Center, Developmental Studies Hybridoma Bank, Korea Drosophila Resource Center, and Kyoto Drosophila Stock Center for reagents; Kenneth Wilson and Pankaj Kapahi for raw data from their phenotypic DGRP screens. This work was supported by grants from the Suh Kyungbae Foundation (SUHF-17020101[CL]); from the National Research Foundation funded by the Ministry of Science and Information & Communication Technology (MSIT), Republic of Korea (NRF-2021M3A9G8022960 [CL]; NRF-2018R1A5A1024261 [CL]; NRF-2023R1A2C100627511 [TK]); from Basic Science Research Program through the National Research Foundation funded by Ministry of Education (NRF-2018R1A6A1A03025810 [TK]).

## Additional information

### Funding

| Funder | Grant reference number | Author |
|---|---|---|
| Suh Kyungbae Foundation | SUHF-17020101 | Chunghun Lim |
| National Research Foundation of Korea | NRF-2021M3A9G8022960 | Chunghun Lim |
| National Research Foundation of Korea | NRF-2018R1A5A1024261 | Chunghun Lim |
| National Research Foundation of Korea | NRF-2023R1A2C100627511 | Taejoon Kwon |
| National Research Foundation of Korea | NRF-2018R1A6A1A03025810 | Taejoon Kwon |

The funders had no role in study design, data collection and interpretation, or the decision to submit the work for publication.

### Author contributions

Jiwon Jeong, Kujin Kwon, Conceptualization, Formal analysis, Validation, Investigation, Visualization, Methodology, Writing – original draft; Terezia Klaudia Geisseova, Jongbin Lee, Investigation; Taejoon Kwon, Chunghun Lim, Conceptualization, Formal analysis, Supervision, Funding acquisition, Visualization, Writing – original draft, Writing – review and editing

### Author ORCIDs

Jiwon Jeong ⬚ http://orcid.org/0009-0006-3111-2656
Kujin Kwon ⬚ http://orcid.org/0009-0004-6642-222X
Jongbin Lee ⬚ https://orcid.org/0000-0002-5868-7437
Taejoon Kwon ⬚ https://orcid.org/0000-0002-9794-6112
Chunghun Lim ⬚ https://orcid.org/0000-0001-8473-9272

### Decision letter and Author response

Decision letter https://doi.org/10.7554/eLife.103973.sa1
Author response https://doi.org/10.7554/eLife.103973.sa2

## Additional files

### Supplementary files
MDAR checklist

## Data availability

The datasets generated and analyzed during the current study are included in source data files or available in the European Nucleotide Archive repository (accession number PRJEB61423). The python scripts that support the findings of this study are available from the author's GitHub webpage under the links https://github.com/taejoonlab/tracking-fly, (copy archived at *Jeong et al., 2021a*) and https://github.com/KJKwon/2023_FlyBehavior, (copy archived at *Jeong et al., 2023*).

The following dataset was generated:

| Author(s) | Year | Dataset title | Dataset URL | Database and Identifier |
| --- | --- | --- | --- | --- |
| Lim C | 2023 | Gene expression analysis of 6 DGRP lines based on clustering behavior and housing | https://www.ebi.ac.uk/ena/browser/view/PRJEB61423 | EBI European Nucleotide Archive, PRJEB61423 |

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
