## [Editor Report]

This study presents important findings on the role of Drosulfakinin signaling in encoding early-life social memory in *Drosophila*, which influences adaptive social plasticity in adulthood. The research demonstrates the neurogenetic basis of social clustering and behavioral adaptation, advancing our understanding of social behavior's molecular and evolutionary bases. The evidence is solid, given the robust genetic, behavioral, and transcriptomic analyses.

---

## [Decision Letter]

[Editors' note: this paper was reviewed by Review Commons.]

---

## [Author Response]

*General Statements [optional]*

Our original manuscript entitled, “Drosulfakinin signaling encodes early-life memory for adaptive social plasticity” (manuscript number: RC-2024-02466R), was reviewed by four reviewers via the Review Commons, and we are now transferring the fully revised manuscript to *eLife* for your consideration. As you will see in our point-by-point responses below, the review was generally favorable, requiring additional data supplemental to our main conclusions, better clarification of method details, and statistical justifications. Please refer to the summary of our biological questions and key findings in the manuscript.

Question: animal species display differential preferences for social networks among conspecific individuals, and social interactions can further impact physiological properties in individuals. Nonetheless, it remains elusive how group properties have evolved with individual traits; and how animals process social experience to shape their physiology. We employ *Drosophila* as a genetic model for social network behavior (SNB) to address these questions.

Key findings: our approaches align large-scale datasets of *Drosophila* behaviors in groups to physiological traits, differential gene expression, and neurogenetic manipulations and unveil novel principles of SNB and the underlying mechanisms:

*Drosophila* social interactions provide a compensatory mechanism for inferior individual traits during development and in adult physiology.

*Drosophila* SNB shows robust plasticity depending on early-life social experience and physiological challenges, and the adaptive “social plasticity” is actually accompanied by conserved genetic reprogramming.

The neuropeptide signaling from pairs of brain neurons expressing Drosulfakinin (DSK) serves as a neural substrate for early-life “social memory” that persists throughout development and supports SNB plasticity.

Given the important advances of significance our work made, we believe that our revised manuscript compares favorably with the high-quality research published by *eLife* and will be of keen interest to the journal’s broad readership, including those interested in the neural basis of social behaviors and their evolution. I hope you find our revised manuscript ready for publication in *eLife*.

*Point-by-point description of the revisions*

Please find our point-by-point responses in blue to the reviewers' comments below.

Reviewer #1 (Evidence, reproducibility and clarity (Required)):The manuscript by Jeong et al. describes effects of neuronal signalling on collective behavior by measuring social distance (SD) which is used as a measure for social network behavior. Authors screened for a panel of inbred DGRP lines and compared the SD due to prior experience of group or single culturing when flies are recorded in a 55 mm diameter petri-dish. The screen uncovered 3 short Sd and three long-SD lines, and subsequent experiments showed differences in various behaviors such as recovery from injury, search for food and SD. Using RNA-seq from heads of flies they implicate Dsk signalling and show neuronal architecture and activity differences between grouped and isolated male flies. They implicate Dsk signalling in recovery from injury affecting SD but it was dispensable for grouped vs. isolated flies. I have suggestions to support the claims made, analysis and interpretation of the data and improve the clarity of writing. See my specific comments below.Major comments:1. For recording social behaviour in flies, arenas with sloped walls have been extensively used called 'fly bowl' (Simon et al. 2010 doi:10.1371/journal.pone.0008793; Robie et al., 2017, doi: 10.1016/j.cell.2017.06.032), or 'flyworld' (Liu et al., 2018 doi:10.1371/journal.pcbi.1006410). Such geometry ensures that flies don't walk on the side of the arena, and don't occlude each other. However, in the screen carried out in this manuscript, a petri-dish of 5.5 cm diameter filled with agar was used to record social network formation. Given the propensity of flies to walk on the walls of such circular arena, it will be difficult to know if the long and short SD behavior resulting from propensity to form clusters is an artefact of the assay condition used. It would be important to test the SNB and SD of at least the 6 selected short and long SD lines in arena with sloped walls to rule out this possibility.

To minimize complications from side-wall walking or z-stacking of individual flies, the circular arena used in our study was filled with agar, making a space of 5.5 cm in diameter x ~0.15 cm in height (see Author response image 1). In fact, the surface tension of the agar solution led to a rounded interface between the agar bed and the side wall that could prevent side-wall walking and partially mimic the sloped wall effects. We included representative video clips for SNB recording in the revised manuscript to further justify our experimental conditions (Video S1-S10). We also revised our method text accordingly and cited the relevant reference.

**Author response image 1. sa2fig1:** The circular arena for SNB recording.

2. Methods section would require additional details about the SNB assay for instance, the height of the agar bed and the effective height in which interactions was recorded is not mentioned.

As described in our response to reviewer #1, major comment #1 above, we revised our method text accordingly.

3. Figure 2C and 2D results in larvae seems to contradict previous studies that have shown that isolated flies eat more as adults (Li et al. Nature, 2021) and Dsk-RNAi increases feeding in larvae and adults (Soderberg et al., 2012). It might be due to unique characteristic of DGRP lines used and would be helpful to discuss this.

We reason that high-digging activity in a group of individual larvae can increase the accessibility to “solid” food, thereby promoting their food intake over 12-h during development. However, food consumption rates and their regulation can vary depending on developmental stages or feeding conditions (e.g., larvae vs. adults; liquid vs. solid food; long-term vs. short-term) (https://pubmed.ncbi.nlm.nih.gov/30914005/; https://pubmed.ncbi.nlm.nih.gov/24937262/). It is thus not fair to align our results directly to those observed under very different biological/experimental contexts. For instance, Li et al. measured the amount of liquid food consumption in individually isolated adults from group culture vs. transient social isolation (i.e., 1-week isolation after eclosion). Soderberg et al. assessed the 15-minute feeding activities of larvae or adults on solid food but did not compare the feeding activity between grouped vs. isolated individuals. Therefore, it is not conclusive whether the DSK-depletion phenotypes are relevant to social experience or whether DSK signaling controls short-term feeding per se. Accordingly, we believe our observations do not necessarily contradict previous studies or indicate characteristics unique to the DGRP lines used in our study.

4. Rutabaga mutants for Figure S3 are directly compared with CS flies in the maze assay and it appears from methods that these lines were not isogenized, this can significantly impact the results. Similarly for some of the subsequent Dsk experiments it appears that lines were not isogenized (see below). These experiments would either need to be repeated of this caveat needs to be explicitly mentioned to avoid misinterpretation of the data.

Since genetic backgrounds could substantially contribute to mutant phenotypes, we mentioned the caveat in our revised manuscript. As the reviewer suggested above, testing isogenized lines could be one option to confirm the genetic effects. Our study took an alternative approach where the observed phenotypes were validated by independent genetic models. For instance, the importance of DSK signaling in injury-induced SNB plasticity was validated by genomic deletions of DSK and DSK receptor genes, as well as by transgenic RNAi (Figure 7B and 7C). In the revised manuscript, we examined additional mutant alleles of *rutabaga* (Figure S6, rut[1] and rut[2080]), *CCKLR-17D1* (Figure S15, CCKLR-17D1[delta1] and CCKLR-17D1[delta2]), and *CCKLR-17D3* genes (Figure S15, CCKLR-17D3[delta1] and CCKLR-17D3[delta2]) to substantiate our original findings.

5. For Figure 3 describing RNA-seq data additional analysis would be helpful. Gene expression from isolated and grouped flies have been studied earlier by microarray and RNA-seq methods (Wang et al., PNAS 2008; Agrawal et al., JEB 2020; Li et al. Nature, 2021). Data from these studies should be compared with to see if there are common patterns of gene expression between long and short SD flies vs. group and isolated flies.

According to the reviewer's suggestion, we compared our DEG analysis with those reported in the previous studies. Genes upregulated in socially deprived flies overlapped substantially between our data and the published ones. However, the number of genes commonly upregulated in grouped cultures was very limited in the pairwise comparisons, and *Dsk* was the only gene upregulated across DEG analyses. Also, DEGs between the short vs. long SD lines barely overlapped with those between grouped vs. isolated flies across independent studies. We speculate that *Drosophila* has evolved a genetic reprogram where social isolation robustly induces the expression of select genes regardless of genetic backgrounds (i.e., DGRP lines in our study vs. Canton-S in the previous studies), whereas diverse genetic pathways shape the baseline SD traits. We revised our text accordingly and included these new analyses in the revised manuscript (Figure S10 and Dataset S3).

6. GEO accession number and the analyzed list of DEGs should be provided as supplementary information.

As described in our original manuscript, we submitted our raw data to the European Nucleotide Archive (ENA accession number PRJEB61423). Since ENA and GEO share their data, uploading our data redundantly onto the GEO should not be necessary. Our original manuscript also included all the DEG lists as supplementary tables (Dataset S1-S3).

7. Figure 3E & F are not referred to in the main text, also there is no description of how the data was generated. Is this based on published data from Mackay lab about DGRP lines, if so, aggression experiments were not convincing in those studies and have been shown to not recapitulate 'real' aggression by other labs for several of the DGRP lines tested (Chowdhury et al., 2021, doi: 10.1038/s42003-020-01617-6).

The main text of our original manuscript actually referred to Figure 3E and 3F. We revised our figure legends to indicate the resources of raw DGRP data and clarified the method for the correlation comparisons. Since we employed the published aggression data from the Mackay lab study (https://pubmed.ncbi.nlm.nih.gov/26100892), we experimentally validated that the long-SD lines indeed show more lunges (i.e., a well-established indicator of aggression behaviors) than the short-SD lines in our revised manuscript (Figure S11).

8. Dsk was shown to be reduced in isolated flies by RNA-seq and play a role in aggression by an earlier study (Agrawal et al., 2020) and should be cited appropriately (line 180-181) and elsewhere.

The paper was appropriately cited in our original/revised manuscript.

9. For Figure 4A-B, source images for other two DGRP lines should be included at least in supplementary information, if not as main figure.

Representative confocal images for the other DGRP lines were included in the revised manuscript (Figure 6A).

10. For Figure 5, what is the reason that uninjured flies don't show any SD phenotype? Are there any changes in their velocity? This is mentioned in passing on line 228-29 but should be properly discussed.

Genomic deletions or transgenic manipulations of the DSK-CCKLR-17D1 pathway gave consistent effects on the injury-induced clustering but not on baseline SD or walking speed. We reason that the DSK-CCKLR-17D1 pathway is dedicated to encoding early-life social experience by enforcing DSK neuron activity and their male-specific postsynaptic signaling. We clarified our text including the genetic background issue in the revised manuscript. Please also see our response to reviewer #1, major comment #4 above.

11. Trans-Tango and UAS-Denmark, SytGFP experiments were performed previously by Wu et al., 2020 and Wang et al., 2021 for Dsk, these two studies observed that P1 neurons are presynaptic and Dsk neurons are post synaptic but in Figure 4 it's not clear what are the presynaptic and post synaptic neurons. Also these studies are not cited appropriately in this section.

The two studies expressed trans-Tango in P1 neurons (P1>trans-Tango) to demonstrate that DSK-expressing neurons are postsynaptic to P1 neurons. They further visualized some overlaps between axon terminals of P1 neurons (P1>sytGFP) and dendrites of DSK neurons (Dsk>DenMark). On the other hand, we expressed the trans-Tango in DSK neurons (Dsk>trans-Tango) to visualize their male-specific/social experience-dependent postsynaptic targets. We also visualized brain regions positive for both synaptic signals from Dsk>sytGFP and the postsynaptic signals from Dsk>trans-Tango. The two studies were cited in our original manuscript to discuss presynaptic partners of DSK neurons and their distinct roles in animal behaviors. We further cited the two studies in this result section of our revised manuscript.

Minor comments:1. Line no. 286: please mention about the relative humidity and light & dark cycle conditions and when experiments were conducted (ZT).

Flies were reared at 40-50% humidity under 12-h light: 12-h dark cycles. Behavioral experiments were primarily conducted between ZT4 and ZT8. We revised the method text accordingly.

2. Line no. 311: How many days old flies were used (isolated and group housed) for the behavior and transcriptomic studies?

We revised the method text to better describe our experimental conditions.

3. Line no. 349: for RNA extraction please mention how many fly heads were used and ZT for collection.

Flies were harvested at ZT4-6, and total RNAs were extracted from 35 fly heads. We revised the method text accordingly.

4. Line no. 358: Italicize "*Drosophila melanogaster*".

Corrected.

Reviewer #1 (Significance (Required)):This manuscript will be of interest to neuroscientists studying *Drosophila* social behaviors. The manuscript asks interesting questions and authors have done extensive set of experiments but the progress appears incremental given the current state of the field, especially for the later part of the manuscript. Some of the interpretation would also require additional data to bolster the claims made. Finally, the findings from this study could be better discussed in the context of what it is already known.

We believe our revised text and additional data in the revised manuscript clarify the reviewer concerns and better support our original findings.

Reviewer #2 (Evidence, reproducibility and clarity (Required)):The article explores the social network behavior (SNB) of *Drosophila*, focusing on individual social distance (SD) within groups over time. A systemic analysis revealed that short SD is associated with long developmental time, low food intake, and hypoactivity. Group culturing compensates for developmental inferiority in short social distance individuals. Social interactions during early development positively impact adult physiology and adaptive social plasticity. Transcriptome analyses show genetic diversity for SD traits. The neuropeptide Drosulfakinin (DSK) signaling mediates social network behavior plasticity via receptor CCKLR-17D1, particularly in males, suggesting a dedicated neural mechanism encoding early-life experiences to adaptively transform group properties. The research suggests that animals have developed neural mechanisms to encode early-life experiences. It offers insights into the genetic foundation and adaptability of social behavior in Drosophila, shedding light on the neural processes involved in social memory and the adaptive behaviors of groups. These findings have broader implications for understanding similar neural mechanisms governing social memory and group behaviors in other species.Major concerns:Major 1. In Figure 2H, the latency to 75% arrival of short-SD isolated fruit flies (no matter with or without pioneers) is close to that of group fruit flies with pioneers while the latency to 75% arrival of long-SD isolated fruit flies (no matter with or without pioneers) is close to that of group fruit flies without pioneers; How to explain the difference in latency to 75% array between long-SD and short-SD isolated fruit flies? It seems that only in the long-SD fruit flies from the grouped experience, the absence of pioneers will increase the time it takes to reach the target food in the maze.

Social deprivation effects on pioneer-free group foraging somewhat varied across the long SD lines (Figure 4B and S5, blue). We reason that the hyperactivity of individual long-SD flies facilitates their food-seeking behaviors in the maze, weakening the pioneer effects or even overriding the group property. Nonetheless, our statistical analyses validated that (1) prior social experience did not significantly affect the group performance of 16 naive flies in the maze assay (the only exception was DGRP707, a long-SD line that showed longer latency in group-cultured naive flies than in socially isolated ones), (2) the presence of pioneers significantly shortened the latency in both short- and long-SD lines, and (3) the pioneer effects were evident only in group-cultured flies. We accordingly revised our result text to better elucidate our conclusion.

Major 2. In Figure 3C, there are two up-regulated genes in the *Drosophila* group that overlap in short-SD and long-SD strains. Apart from Dsk, what is the other gene? In addition, for isolated fruit flies, both short-SD and long-SD lines have more gene expression upregulated. How to explain this phenomenon? Can you briefly explore the reasons for their upregulation and instead of involvement in SNB plasticity, what kind of physiological functions may they have?

The other gene commonly upregulated in group-cultured DGRP flies was Arc1 (Activity-regulated cytoskeleton-associated protein), implicated in synaptic plasticity and fat metabolism (https://flybase.org/reports/FBgn0033926.htm). Arc1 downregulation upon social isolation could be relevant to the weak postsynaptic signaling of DSK neurons (Figure 6D and 6E) or be a part of the metabolic reprogramming (Figure 5D; also see below). Nonetheless, we focused on the neuropeptide DSK, given its unique expression in the brain and relevance to other social behaviors (e.g., mating, aggression). In fact, *Dsk* was the only overlapping gene that was downregulated upon social isolation across independent studies (Figure S10A).

As the reviewer pointed out, social isolation upregulated many genes, including those involved in metabolism. Our revised manuscript additionally showed that upregulated but not downregulated genes upon social isolation were substantially conserved across genetic backgrounds or independent DEG studies (Figure 5C and S10A). We speculate that *Drosophila* has evolved a genetic reprogram where social isolation elevates metabolic gene expression to adaptively induce a metabolic shift for energy storage and fitness. We revised our text accordingly.

Major 3. In lines 219-223, the genomic deletion by mutant or depletion by RNA interference emphasizes the role of neuropeptides DSK and its receptor CCKLR-17D1 in injury-induced clustering behaviors. How about the effect of neuropeptides overexpression? Do they confer injury-induced social interactions to isolated male flies. Meanwhile, in line 238, the transgenic excitation of CCKLR-17D1 neurons emphasizes the function of neuronal synaptic transmission in the pathway. Indeed, both neuropeptide expression and neuronal synaptic connections may be involved in the regulation of injury-induced clustering behaviors. It is recommended to separate the discussion of protein expression and the respective regulatory modes at the neuronal circuit level.

We could not test DSK overexpression effects on injury-induced clustering in socially isolated males since we failed to validate DSK overexpression from a relevant transgenic line (https://flybase.org/reports/FBal0184043.htm). Instead, we provided additional data in the revised manuscript that independent genomic deletions of the CCKLR-17D1 locus (Figure S15) or transgenic silencing of the synaptic transmission in CCKLR-17D1 neurons (Figure S16) suppressed the injury-induced clustering in group-cultured male flies. According to the reviewer's suggestion, we modified our text to better distinguish between the effects of gene/protein expression vs. relevant neuron activities on social behavior plasticity.

Major 4. Since a significant portion of the work in the first half of this paper is focused on elucidating two types of social distance in SNB, is there any difference in the regulation of social network plasticity by Dsk signaling pathway in the short-SD and long-SD lines?

As the reviewer suggested, it will be informative to determine if Dsk signaling for social behavior plasticity is differentially regulated in short- vs. long-SD lines. One technical issue is that genetic factors shaping their SD traits still need to be defined. So, we are limited to performing standard genetic/transgenic experiments using the DGRP lines while retaining their SD phenotypes. Accordingly, our current approach was to compare DSK expression in short- vs. long-SD lines under grouped- vs. isolated-culture conditions. Future studies should address the review comment above.

Minor ones:Minor 1. There is a color difference between the data spots and the figure legends in Figure 2H.

Corrected.

Minor 2. The anatomical sample images in Figure 4 and Figure 5 require scale bars.

We added scale bars to Figure 6 and 7 in the revised manuscript.

Minor 3. The "grouped" and "grp" in Figures 3B-3F can be unified as "grp", while the "isolated" and "iso" can be unified as "iso". So that the male and female symbols in Figure 3F will not have any deviation in the mark.

We unified the labels throughout the revised manuscript according to the reviewer's suggestion.

Minor 4. The difference in Denmark signals of each group of neurons under the condition of injury should also be compared in Figure 4C.

The DenMark signals were also compared between control and injury conditions (Figure S12).

Minor 5. What is the effect of inactivating CCKLR-17D1 or CCKLR-17D3 by shibire on injury-induced clustering in group-cultured adults in Figure 5E? (This relates to major comment 3)

We actually employed a tetanus toxin light chain (TNT) to block synaptic transmission in CCKLR-17D1 neurons and found that the transgenic manipulation of CCKLR-17D1 neuron activity suppressed injury-induced clustering in group-cultured males (Figure S16). Since (1) our additional data using independent deletion alleles further excluded the possible implication of CCKLR-17D3 in the SNB plasticity (Figure S15) and (2) a transgenic Gal4 knock-in for the CCKLR-17D-3 locus is not available, we focused on the CCKLR-17D1 experiments in our current study and wished to leave more detailed circuit analyses for future studies.

Reviewer #2 (Significance (Required)):

General assessment:The strengths of this work is that the authors have identified specific lines with short social distance or long social distance by conducting extensive screening experiments. By transcriptome analyses and gene ontology (GO) analyses they revealed genes up or down regulation in the social experience. They have also narrowed down to the DSK signaling involved in the social experience encoding process. However, the study's limitation lies in the lack of clarity regarding the DSK signaling pathway. The mechanisms through which social experiences affect neuronal activity and synaptic connections remain unclear. Further research on upstream and downstream pathways could enhance understanding. Although the article proposes injury-induced clustering behaviors, the key sensory pathways involved in social network behavior plasticity during early social experiences are not well-defined. Conducting sensory deprivation experiments could elucidate sensory involvement. Overall, the study's strengths lie in its comprehensive approach, large sample size, and translational potential. To enhance future research, investigating the complexity of neural mechanisms and expanding the exploration of regulating pathways could be beneficial. Additionally, exploring the ecological relevance of the findings could deepen our understanding of social behavior in natural environments.

Our current work provides a neuroanatomical basis for early-life social memory and experience-dependent plasticity of social-interaction behaviors. We believe future studies will build up the mechanical details for social experience-dependent DSK expression, DSK neuron activity, and behavioral outputs. Regarding the key sensory pathways, we examined injury-induced SNB plasticity of distinct sensory mutants (e.g., olfactory, visual, auditory, etc.) and our revised manuscript provided additional data that norpA-dependent visual sensing might play a crucial role in this process (Figure S9), consistent with the previous finding that vision is required for larval clustering behaviors in *Drosophila* (https://pubmed.ncbi.nlm.nih.gov/28918946/).

Advance:Compared to previous studies such as Heiko Dankert et al.'s publication in 2009 in Nature Methods and Assa Bentzur et al.'s publication in 2020 in Current Biology, which also investigated the impact of early life experiences on male social behavior and examined various aspects of social network construction, this study employs a systematic analysis of social network behavior (SNB) in *Drosophila*, integrating genetic, physiological, and behavioral assessments. The authors conducted detailed and systematic analyses through transcriptome and gene ontology (GO) analyses, including the visualization of gene expression heatmaps, volcano plots, and overlapping analysis of differentially expressed genes (DEGs) between grouped and isolated conditions. Additionally, this research delved into the regulatory pathway of DSK signaling in male-specific SNB plasticity, with a particular focus on the DSK to CCKLR-17D1 signaling, which encodes early social experiences. The research provides valuable insights into the genetic basis and adaptability of social behavior in Drosophila. Moreover, it illuminates the neural mechanisms that underlie social memory and the ability of groups to adapt across different species.Audience:Researchers conducting basic research in genetics, neuroscience, behavioral biology, and evolutionary biology, particularly those focused on understanding social behavior and its underlying genetic and neural mechanisms, will find this study highly relevant. Additionally, researchers studying social cognition, social memory, and group dynamics in various species may also be interested in these findings.Reviewer #3 (Evidence, reproducibility and clarity (Required)):Jeong et al. investigate the influence of genetic factors and early-life social experience on social network behaviors in adult *Drosophila*. Utilizing isogenic DGRP lines, the study correlates social distances with key developmental and physiological traits-developmental time, digging activity, and food intake. The findings suggest that adult flies with shorter social distance -indicating closer proximity to each other-face developmental disadvantages that are offset by the benefits of social grouping. The authors argue for an evolutionary advantage in such social behaviors, suggesting they help compensate for individual developmental deficits. The study further identifies the Dsk signaling pathway as a key mediator of social network behavior plasticity in male flies, particularly under challenging conditions like mechanical injury.The study undertakes a broad range of behavioral and neurogenetic approaches, demonstrating an extensive scope of research efforts. Despite its ambitious scope, the manuscript lacks a clear rationale and cohesive flow among its sections. The numerous experiments do not merge into a unified narrative, leaving the reader questioning the reasoning and progression behind the experimental choices. The manuscript needs a clearer structure, well-defined hypotheses, and more detailed methodological descriptions. Greater emphasis on novelty and better integration with existing literature are also needed. The lack of control experiments and adequate statistical analysis weakens some conclusions.Major comments1. The authors show that flies in short SD lines reduce their activity over time, leading to the formation of social clusters (Figure 1B). This clustering could potentially be attributed to reduced activity rather than active social preferences. It would be informative to test whether these SD flies exhibit similar social behaviors when placed in a larger arena, to test if the clustering persists under varied environmental conditions.

Short-SD flies did not reduce their moving speed over time when we placed a single fly in the original arena and assessed its locomotor behavior individually (Figure S2). Thus, it is likely that the reduced activity in a group of short-SD flies is an effect of clustering over time but not necessarily the cause. We also confirmed that short and long SD lines retain their clustering property even in a larger arena (8.5 cm in diameter) (Figure S3). We included these new data in our revised manuscript to better demonstrate active social preferences in the DGRP lines.

2. In lines 78-79, the authors claim that "short-SD flies gradually reduced SD over time and stayed in the cluster." However, the study established SD clustering by only analyzing behavior during the last quarter of a 10-minute window, assigning a single data point to each fly and taking the average for group SD. Yet, a single value cannot demonstrate whether initially formed clusters remained stable-unchanged-over time. To strengthen this point, the authors could investigate dynamic changes in SD over a longer period to demonstrate stability, or alternatively, adjust the language to better convey the findings. Additionally, including a time scale in Figure 1B would enhance the clarity of these findings.

We traced dynamic changes in SD and walking speed of representative DGRP lines over the 10-minute window (Figure 2A and 2B) and modified our text accordingly in the revised manuscript. We also included a time scale in Figure 1B and relevant figures.

3. The statistical analysis presented in Figure 2C-D raises concerns. It appears that feeding and digging efficiency in both SD type lines benefit from socialization, suggesting that the effects attributed to SD might stem from the overall digging and feeding activity of each line. Therefore, it is crucial to integrate both social distance (short vs. long) and socialization (grouped vs. isolated) into the analysis using methods that allow for the assessment of confounding effects (interaction), such as two-way ANOVA or regression, depending on the data. This would help authors to clarify whether isolation reduces feeding overall (both line types) and determine if this reduction is more pronounced in short-SD lines. Additionally, it is counterintuitive that lines with more larvae per cluster show worse digging efficiency when previous studies, such as Dombrovski et al. (2017), have shown that larger groups of larvae typically exhibit better digging efficiency. This discrepancy highlights the need for a thorough re-evaluation of the data and assumptions regarding group dynamics and their impact on resource access.

As the reviewer suggested, we employed ordinary or aligned ranks transformation 2-way ANOVA (depending on the normality and equal variance of a given dataset) to determine if social distance and socialization cooperatively contribute to developmental phenotypes. Our new analyses confirmed significant interaction effects of social distance and socialization on most developmental phenotypes tested (i.e., larval digging activity, developmental time, %male progeny, and %eclosion success). These results convincingly support that short-SD larvae benefit more from socialization than long-SD larvae to compensate for the inferior phenotypes in isolated individuals. We speculate that the feeding amount of isolated long-SD individuals may be saturating for normal development (i.e., developmental time, %male progeny), possibly explaining the lack of interaction effects on food intake while displaying developmental inferiorities only in isolated short-SD individuals. We reason that grouped long-SD flies should not necessarily display poorer digging activity than grouped short-SD flies since isolated long-SD individuals displayed much higher digging activity than isolated short-SD individuals. Consistent with the previous finding, both SD lines showed better digging efficiency when grouped than isolated. We included these new analyses in the revised manuscript and modified our text accordingly. To clarify any statistical issues, we included a summary of all our statistical analyses performed in the revised manuscript (Dataset S5).

4. The choice to use the percentage of male progeny as a measure of developmental success is confusing, especially without an explanation for why it is favored over measures like overall progeny survival rates. As with digging and feeding, the statistical analysis should include an examination of potential interaction effects to fully assess how social conditions impact developmental outcomes.

The percentage of male progeny was one of the most evident developmental phenotypes on which social distance and socialization showed significant interaction effects. In the revised manuscript, we further included the percentage of eclosed flies as a measure for the overall progeny survival rate (Figure 3F) and performed 2-way analyses to validate the significant interaction effects of social distance and socialization on various larval/developmental phenotypes. Please see our response to reviewer #3, major comment #3 above.

5. The rationale for using physical injury to induce SNB in the study is not clearly explained, raising concerns about the potential impact of injury on overall locomotion. Before employing such a method in sociality experiments, it is crucial to demonstrate that the injury does not affect locomotion. Additionally, the study's methodologies for transitioning between grouped and isolated cultures (present only in Figure 2I and not in the methods section), as well as the specific methods used to measure social distance (SD) in isolated flies, are not sufficiently detailed. This lack of clarity complicates the evaluation of the study's conclusions.

To determine if the long-SD lines express their social behaviors selectively (e.g., upon physiological challenges), we introduced physical injury to the SNB analysis. There was a positive correlation between locomotion activity and SD trait among DGRP lines (i.e., DGRP lines with low walking speed and centroid velocity exhibited short-SD phenotypes in general) (Figure 1D). This observation thus prompted us to hypothesize that modest injury may reduce locomotor activity in individuals, facilitate their interactions in a group, and shorten the overall SD. The mechanical injury actually shortened SD in both the short- and long-SD lines (Figure 4D and S7A). Under the same experimental condition, mechanical injury reduced walking speed and centroid velocity only in the long-SD lines (Figure S8), whereas social isolation blunted the injury effects (Figure 4D, S7A, and S8). We reason that our injury condition does not severely impair general locomotion per se to abolish or overestimate SNB, but low activity in grouped long-SD flies is likely a consequence of their injury-induced clustering. We clarified our original text for the rationale and included the new data in the revised manuscript (Figure S8). We also revised the method text for the transitions between grouped and isolated cultures, as well as for measuring SD in isolated flies.

6. Lines 104-106 "The clustering property of short-SD lines may have evolved as a compensation mechanism for their developmental inferiority in individuals". To support this claim, the authors should assess the significance of interactions terms as stated earlier.

Please see our responses to the reviewer’s relevant comments above (reviewer #3, major comments #3 and #4).

7. In Figure 4, the authors conclude that Drosulfakinin (DSK) signaling encodes early-life experiences for SNB plasticity. It is crucial for the authors to differentiate whether changes in feeding behavior are directly due to DSK or if they are secondary effects resulting from altered social interactions mediated by DSK.

Previous studies demonstrated that DSK is a satiety-signaling molecule whose expression is elevated upon feeding to suppress food intake (https://pubmed.ncbi.nlm.nih.gov/34398892/; https://pubmed.ncbi.nlm.nih.gov/32314736/; https://pubmed.ncbi.nlm.nih.gov/25187989/; https://pubmed.ncbi.nlm.nih.gov/22969751/). Under our experimental context, social isolation downregulated DSK expression and DSK neuron activity, whereas isolated larvae rather reduced their food intake. It is thus unlikely that changes in the feeding behavior of isolated larvae directly implicate DSK-dependent satiety signaling. We discussed this issue in our revised manuscript.

8. In Figure 4, the data show that DSK peptide is significantly increased in cell bodies in grouped long DS lines when compared with grouped short DS lines (Figure 4B). However, no changes are reported at the level of DSK projection levels when comparing these groups. Can the authors clarify this?

The SD-trait effects on DSK levels were evident in cell bodies but not in DSK neuron projections. We reason that axonal transport or processing of the neuropeptide was limiting under the group-culture condition. These observations might also be relevant to our conclusion that *Dsk* is not crucial for shaping the SD traits per se. We revised our text accordingly.

9. Additionally, the data show that DSK activity is reduced by isolation in both types of SD. To clarify if this effect is driven by isolation only, and not type of line (short vs long SD line), the interaction term should be tested. Furthermore, it is not clear what lines are used in live imaging (e.g., Figure 4C-F).

We detected no significant interaction effects of SD type and social isolation on DSK expression (Figure 6B). Live-brain imaging of the GCaMP-expressing DSK neurons was performed using a transgenic line (i.e., Dsk-Gal4>UAS-GCaMP) in a wild-type background. Since genetic factors shaping the SD traits were not defined in each DGRP line, we could not combine the transgenes with DGRP backgrounds while retaining their respective SD phenotypes (please also see our response to reviewer #2 major comment #4 above). Nonetheless, the GCaMP imaging demonstrates that (1) either injury or social isolation alone significantly affects DSK neuron activity, but (2) the two conditions act independently on the GCaMP signals (i.e., no significant interaction effects). We clarified it in our revised manuscript and displayed each genotype used in our imaging experiments.

10. In the 'Male-specific DSK-CCKLR-17D1 signalling mediates SNB plasticity' section (line 217), the analysis should include an interaction term to account for the possible confounding effects of isolation and injury on SD. This would aid in determining whether the impacts of social isolation and injury on DSK signalling and SNB plasticity are independent of each other or if they interact in significant ways, as stated by the authors.

Throughout our revised manuscript, we performed 2-way analyses to validate the interaction effects of isolation and injury on SD and support our conclusion. We also included a summary of all our statistical analyses performed in the revised manuscript (Dataset S5).

Minor comments1. The introduction section would also benefit from major rewriting to clearly indicate the research gap and hypothesis tested. The introduction section would also benefit from major rewriting to clearly indicate the research gap and hypothesis tested. The manuscript would benefit from a more thorough integration of previous studies related to *Drosophila* social behavior (e.g., Blumstein, D. T. et al., 2010; Schneider, J., Dickinson, M. H., & Levine, J. D., 2012; Simon, A. F. et al., 2012; Ramdya, P. et al., 2015). While the current references are adequate, a more detailed discussion of how this study builds upon and diverges from existing literature would be beneficial.

The introduction of our original manuscript starts from previous findings on *Drosophila* social behaviors and clearly indicates what remains elusive, thereby defining our biological questions. We further explain why we focus on SD among other social network measures published previously and outline our approaches for new findings in this study (i.e., the principles of social network behavior and its plasticity). Since our original text was written in a concise manner, we revised our text in both the introduction and result sections to give a more detailed description of what earlier studies have discovered according to the reviewer suggestion.

2. A better description of methods, especially behavioral approaches, could vastly help in understanding the results. Clarifying the methodologiy, particularly the behavioral approaches, would greatly enhance the understanding of the results. Also, the method for quantifying the total number of larvae per vial is unclear, particularly whether variations in larval density were considered. This is crucial, as different densities could affect the available sensory cues necessary for larval aggregation, such as vision (e.g. Dombrovski et al. Curr Biol. 2019). Better descriptions of the results and inclusion of exact statistical analyses used in support of the claims are also needed.

We revised our method text to better describe our experimental conditions. We further described how we controlled larval density to prepare group-cultured larvae and adults for analyzing larval behaviors and developmental phenotypes. Finally, we included a summary of all our statistical analyses performed in the revised manuscript (Dataset S5).

3. Some terms and descriptions in the manuscript are somewhat ambiguous, such as "social memory" and "adaptive social plasticity" and should be better defined.

We better defined the two key short terms in the introduction of our revised manuscript.

4. Line 86-89: "Social interactions compensate for developmental inferiority in short-SD larvae Why do flies display SNB? One clue comes from the previous observation that *Drosophila* larvae collectively dig culture media and improve food accessibility, possibly facilitating their constitutive feeding during early development…" – This paragraph could be moved to the introduction section.

As we reorganized the introduction in our revised manuscript, we feel it should be fine to leave the paragraph above in the original context.

5. In lines 77-78, the manuscript mentions that the locomotion trajectories of individual flies confirm certain characteristics but fails to provide an analysis of individual locomotion metrics, e.g., tortuosity, distance walked, etc. The authors should add quantitative analysis to support claims about trajectories or alternatively rephrase the sentence to remove any claims about the trajectories of flies.

As the reviewer suggested, we added quantitative analyses of cumulative walking distances over time and total walking distances in individual flies to our revised manuscript (Figure 2D and S1C).

6. After screening 175 strains, three short and long SD lines were selected. It would be good if justification for the authors' choice were included, as the selected lines were not the ones with the longest or shortest SD as seen in Figure 1C.

We ranked individual DGRP lines for each of the two group properties (i.e., SD and centroid velocity) and then selected the top and bottom three lines based on their average ranking. We included this rationale in our revised manuscript.

7. Other comments:• Line 72-73: What correlation was performed? This should be included in the results/methods section.

As described in the figure legend of our original manuscript, the significance of the correlation was determined by Spearman correlation analysis. We further included the method description in the results and methods section of our revised manuscript.

• Line 113: Change "pre-trained colleagues" to "pre-trained flies".

Changed.

• Lines 321 and 325: Use "3D" instead of "2D" as three dimensions are given?

Corrected.

• Ensure all figures are correctly scaled and aligned.

We revised our figures to avoid any of these issues.

• Video: Including short videos for each behavioral test (e.g., feeding) would help in understanding it.

We included representative video files for SNB and aggression assays in the revised manuscript (Video S1-S12).

• Figure 4 should include control neurons that do not change with social grouping; authors should also show ROI.

We included new data for control neurons (Figure S13, *Pdf*-Gal4>UAS-GCaMP7f) and also showed ROI for quantification in our revised manuscript (Figure 6C and S13).

• Line 27, 86: Change "inferiority" to "disadvantage".

We feel inferiority fits better in the context of our overall study.

Reviewer #3 (Significance (Required)):This study extends existing knowledge by linking specific genetic pathways to behavioral outcomes in a well-established model system, providing new insights into the genetic and neural basis of social behavior. The use of DGRP lines to dissect the impact of genetic variation on behavior is particularly valuable. The identification of the Dsk signaling pathway as a mediator of these behaviors under stress is interesting. However, the study would benefit from more in-depth statistical analysis and expanded experimental designs to solidify the conclusions. It should also more clearly highlight the novelty of its findings and better integrate them with the current literature on Dsk signaling and social behaviors.My expertise is in behavioral neuroscience. The insights from this study promise to deepen our understanding of the genetic and neural mechanisms behind social behaviors. The potential implications of this research are likely to extend well beyond *Drosophila*, influencing studies across various species.Reviewer #4 (Evidence, reproducibility and clarity (Required)):Summary:The manuscript investigates the role of Drosulfakinin (Dsk) signaling in *Drosophila* social network behavior (SNB) and its plasticity based on early-life experiences. The study employs a systematic analysis using 175 inbred strains to link short social distance (SD) with developmental time, food intake, and activity levels. Key findings suggest that social interactions during development compensate for individual developmental inferiority and that early-life social experience is necessary for adaptive social behaviors in adults. The genetic basis of SNB is further explored through transcriptome analyses, implicating Dsk and one of its receptors in mediating these behaviors.Major comments:Are the key conclusions convincing?The key conclusions are well-supported by the data presented. The association between early-life social interactions and adult social behaviors is convincingly demonstrated through multiple experimental setups.

We appreciate the reviewer’s positive feedback on our rigorous approaches and key conclusions.

Should the authors qualify some of their claims as preliminary or speculative, or remove them altogether?Some claims, particularly those regarding the evolutionary implications of Dsk signaling and its conservation across species, might benefit from being presented as hypotheses or speculations rather than definitive conclusions. This would align with the current evidence while acknowledging the need for further investigation.

As the reviewer suggested, we toned down our claims on the evolutionary implications of Dsk signaling.

Would additional experiments be essential to support the claims of the paper?Measure aggression and male-male courtship behavior in the 6 DGRP lines to examine whether SD correlates with these behaviors.

Aggression but not male-male courtship behaviors correlated with SD phenotypes in the 6 DGRP lines. We included the new data in our revised manuscript (Figure S4 and S11). Please also see our response to a relevant reviewer comment above (reviewer #1, major comment #7).

Include behavior results of flies tested in Figures 4C and D.

We included the behavior data in our revised manuscript (Figure S12A).

Repeat the CCKLR-17D1 experiments shown in Figures 5 F and G for CCKLR-17D3 to provide extra evidence that CCKLR-17D1 mediates DSK's effects on SNB.

We employed a transgenic Gal4 knock-in for the CCKLR-17D1 locus to specifically manipulate the activity of CCKLR-17D1-expressing neurons. However, a Gal4 knock-in line for the CCKLR-17D3 locus was not available for pairwise comparison. We instead provide extra evidence for CCKLR-17D1 function in SNB plasticity by showing that (1) independent genomic deletions of CCKLR-17D1 but not CCKLR-17D3 suppressed injury-induced clustering in group-cultured males (Figure S15) and (2) blocking of synaptic transmission in CCKLR-17D1 neurons phenocopied CCKLR-17D1 deletion (Figure S16). Please also see our response to a relevant reviewer comment above (reviewer #2, minor comment #5)

Are the suggested experiments realistic in terms of time and resources?These experiments are realistic and feasible within typical research timelines. These might require a few months and moderate funding.

According to the reviewer suggestions, we included new pieces of data in our revised manuscript to address the reviewer concerns and further support our conclusions.

Are the data and the methods presented in such a way that they can be reproduced?The methods section is detailed, providing sufficient information for replication.

We appreciate the reviewer’s positive feedback on our method description.

Are the experiments adequately replicated and statistical analysis adequate?The experiments appear to be adequately replicated, and the statistical analyses are generally appropriate. However, ensuring consistent selection of statistical methods can further support the evidence presented (Figures 2C and D).

We performed more appropriate statistical analyses in the revised manuscript (e.g., 2-way ANOVA of the data presented in our original Figure 2C and 2D) and included a summary of all the statistical analyses in the revised manuscript (Dataset S5). Please also see our responses to the reviewer comments above (reviewer #3, major comments #3 and #10; reviewer #3, minor comment #2).

Minor comments:Specific experimental issues that are easily addressable:Ensure clarity in the presentation of figures and legends. Some figures could benefit from more detailed legends explaining all aspects of the data shown.

We revised our figures and figure legends to address this issue and improve clarity.

Are prior studies referenced appropriately?The manuscript references prior studies appropriately, providing a solid context for the current research.

We appreciate the reviewer comment.

Are the text and figures clear and accurate?The text is clear, but some figures, particularly those with complex data, could be more informative with additional annotations.

We revised our figures and figure legends to address this issue.

The larvae pictures in Figure 2A should be replaced with ones with higher resolution with drawn larval contours.

We replaced the larval pictures with higher resolution and indicated individual larvae with arrows in the revised manuscript (Figure 3A).

Scale bars are missing in most of the images shown.

We added scale bars to our revised figures.

Do you have suggestions that would help the authors improve the presentation of their data and conclusions?Consider providing a graphical abstract summarizing the key findings. This would aid readers in quickly grasping the main conclusions. Additionally, breaking down complex figures into simpler, more focused panels might improve readability.

As the reviewer suggested, we split complex figures into simpler ones to improve the readability of our revised manuscript and data. We also provided a graphical abstract summarizing our findings (Figure 8).

Reviewer #4 (Significance (Required)):Describe the nature and significance of the advance (e.g. conceptual, technical, clinical) for the field.This study provides significant conceptual advances in understanding the genetic and neurobiological basis of social behavior in *Drosophila*. By linking early-life social experiences to adult social behaviors, it highlights the importance of developmental context in shaping adult phenotypes.Place the work in the context of the existing literature (provide references, where appropriate).The work builds on previous studies on *Drosophila* social behavior and neurogenetics. It extends the current understanding by integrating developmental and adult behaviors with genetic and molecular analyses. References to foundational works in Drosophila social behavior and recent studies on neuropeptide signaling are well-placed.State what audience might be interested in and influenced by the reported findings.Researchers in the fields of neurogenetics, behavioral ecology, developmental biology, and evolutionary biology will find this work particularly relevant. It also has implications for those studying social behavior across species, including mammals.Define your field of expertise with a few keywords to help the authors contextualize your point of view. Indicate if there are any parts of the paper that you do not have sufficient expertise to evaluate.Expertise: Neurogenetics, Behavioral Neuroscience, *Drosophila* Genetics, Social Behavior, Bioinformatics. I have sufficient expertise to evaluate the genetic, behavioral, and transcriptomics aspects of the study. Specific details on the imaging studies might require additional expert evaluation.